

# Evidence for a maximum of sinking velocities of suspended particulate matter in a coastal transition zone

Joeran Maerz[1], Richard Hofmeister[1], Eefke M. van der Lee[1], Ulf Gräwe[2,3], Rolf Riethmüller[1], and Kai W. Wirtz[1]

[1]Institute of Coastal Research, Helmholtz-Zentrum Geesthacht (HZG), Geesthacht, Germany
[2]Leibniz Institute for Baltic Sea Research, Warnemünde, Germany
[3]Institute of Meteorology and Climatology, Leibniz University Hannover, Hannover, Germany

*Correspondence to:* Joeran Maerz (joeran.maerz@hzg.de)

**Abstract.** Marine coastal ecosystem functioning is crucially linked to the transport and fate of suspended particulate matter (SPM). Transport of SPM is, amongst others, controlled by sinking velocity $w_s$. Since $w_s$ of cohesive SPM aggregates varies significantly with size and composition of mineral and organic origin, $w_s$ probably exhibits large spatial variability along gradients of turbulence, SPM concentration and SPM composition. In this study, we retrieved $w_s$ for the German Bight, North Sea, by combining measured vertical turbidity profiles with simulation results for turbulent eddy diffusivity. Analyzed with respect to modeled prevailing energy dissipation rates $\epsilon$, mean $w_s$ were significantly enhanced around $\log_{10}(\epsilon\,(\mathrm{m}^2\,\mathrm{s}^{-3})) \approx -5.5$. This $\epsilon$ region is typically found at water depths of approximately $15\,\mathrm{m}$ to $20\,\mathrm{m}$ on a cross-shore transect. Across this zone, SPM concentration declines drastically towards the offshore and a change in particle composition occurs. This characterizes a transition zone with potentially enhanced vertical fluxes. Our findings contribute to the conceptual understanding of nutrient cycling in the coastal region which is as follows: Previous studies identified an estuarine circulation. Its residual landward-oriented bottom currents are likely loaded with SPM particularly within the transition zone. This retains and traps fine sediments and particulate-bound nutrients in coastal waters where organic components of SPM become re-mineralized. Residual surface currents transport dissolved nutrients towards the off-shore, where they are again consumed by phytoplankton. Algae excrete extracellular polymeric substances which are known to mediate mineral aggregation and thus sedimentation. This probably takes place particularly in the transition zone and completes the coastal nutrient cycle. The efficiency of the transition zone for retention is thus suggested as an important mechanism that underlies the often observed nutrient gradients towards the coast.

## 1 Introduction

Biogeochemical cycling and functioning of marine coastal and shelf sea systems crucially relies on particle transport. Vertical fluxes of suspended particulate matter (SPM) are determined by sinking velocity $w_s$ and indirectly affect the horizontal transport. In coastal systems, SPM is composed of living and non-living particulate organic matter (POM) and fine cohesive and non-cohesive resuspended minerals. Cohesive sediments and POM can undergo aggregation and fragmentation processes that change transport properties and thus their sinking velocity. As a consequence of flocculation, SPM aggregates ubiquitously possess a broad spectrum of size and composition (Fettweis, 2008). This heterogeneity in-between flocs increases the method-





ological effort to analyze $w_s$ *in situ* (Fettweis, 2008). On larger scales, SPM concentration and composition additionally exhibit strong spatio-temporal variability that are the result of manifold interplaying processes. Tidal and wind-induced currents are the major driver for resuspension and subsequent horizontal transport, while biological processes, e. g. algae growth and bio-induced sediment stabilization (Black et al., 2002; Stal, 2003), interfere and thus additionally shape the complex distribution

of SPM in coastal and estuarine systems. Typically, SPM concentration and composition show cross-coastal gradients (Tian et al., 2009; van der Lee et al., 2009; Li et al., 2010). In shallow waters near the coast, where turbulence and thus resuspension are high, SPM concentration is usually enhanced and dominated by mineral particles with high densities. By contrast, in deeper off-shore regions, SPM concentration is comparably low and consists to a higher extent of POM with low densities. This rather general pattern of changing SPM concentration and composition from coast to open waters is also typical for our

research area, the German Bight (Eisma and Kalf, 1987), but was observed worldwide in estuaries (e. g. Fugate and Friedrichs, 2003) and across coastal seas (e. g. van der Lee et al., 2009). Since both, SPM concentration and turbulence, control $w_s$ of cohesive material (Pejrup and Mikkelsen, 2010), it is likely that these cross-coastal SPM gradients induce considerable spatial variability of $w_s$ and thus affect transport and fate of SPM in coastal marine systems. To the authors' knowledge, however, to date no comprehensive analysis has addressed system-wide cross-shore gradients in settling velocity, especially in relation to

possible drivers.

Sinking velocity of SPM is determined by floc size and density, both resulting from a complex interplay of processes. Particle size distribution is locally governed by aggregation, fragmentation and restructuring processes mainly driven by turbulence-induced shear $\overline{G} = (\epsilon/\nu)^{1/2}$ (Pejrup and Mikkelsen, 2010) generated by energy dissipation $\epsilon$ (Camp and Stein, 1943) with $\nu$ being the kinematic viscosity. Under a given shear regime, aggregation is controlled by floc volume concentration, and adhesion

properties of the floc forming primary particles of mineral and organic origin. The higher the volume concentration of flocs, the higher is the encounter rate. Subsequently, adhesion forces of the involved particles eventually determine the probability to stick together. In addition, adhesive forces limit the intrusion of particles during clustering, loosening the floc structure (Meakin and Jullien, 1988). This leads to a floc morphology that shows self-similar fractal scaling (Kranenburg, 1994). As a result, aggregates possess decreased density compared to the comprising primary particles. Adhesion forces between the

particles within a floc, however, strengthen the resistance of aggregates against fragmentation (Kranenburg, 1999), while the smallest eddies, with sizes of the Kolmogorov microscale (Kolmogorov, 1941), potentially limit the maximum particle size (Berhane et al., 1997). In sum, $w_s$ is locally governed by turbulence-driven processes whose rates depend on the physico-chemical properties and volume concentration of particles.

To date, there is still a lack of understanding of how environmental conditions and especially biological processes affect

physico-chemical properties, and thus $w_s$, of cohesive SPM. Typically, a power law relation between median $w_s$ and SPM concentration is postulated and found at local measurements. These relations, however, vary considerably in their power factor among different systems as e. g. summarized by Dyer (1989) for various estuaries. This variation can be attributed to different shear stresses and physico-chemical properties of involved particles. They are particularly subject to algae and microbial extracellular polymeric substances (EPS) excretions, among them transparent exopolymer particles (TEP), that are known

to mediate aggregation processes. TEP bridge and glue mineral particles together (Decho, 1990), and potentially increase





resistance against particle fragmentation (Fettweis et al., 2014). By these mechanisms, TEP are hypothesized to enable phyto-plankton to clear the water column from suspended sediments (Fettweis et al., 2014). Such clearing ability may contribute to spatial variability of $w_s$ which would affect biogeochemical cycles. So far, knowledge on spatial variations of $w_s$ on a system scale is rare. A study on a transect in San Francisco Bay has been carried out (Manning and Schoellhamer, 2013), but gener-

ally, there is no systematic understanding about the relevance of sedimentation variability for biogeochemical cycles in coastal ecosystems. Such knowledge, though, is needed to understand and model coastal shelf biogeochemistry and sedimentology. For example, it is still an ongoing scientific discussion which processes, and to what extent, sustain the net-sediment transport towards the Dutch, German and Danish coast into the Wadden Sea (Postma, 1984). We therefore aimed at a reconstruction and analyses of $w_s$ of SPM for the German Bight, North Sea. We developed a new approach to retrieve $w_s$ from high resolution

turbidity profiles in combination with vertical mixing rates estimated by a hydrodynamical model. Our findings are particularly discussed in the light of their relevance for biogeochemical cycling of matter in coastal waters.

## 2   Methods

In the following, the study area, sampling and preprocessing of observational data are described. Afterwards, the procedure to extract sinking velocities from observations with the aid of hydrodynamical model results is explained.

### 2.1   Study area

The German Bight (Fig. 1) is located in the south-east of the North Sea. The latter is a shallow shelf sea with an average depth of $80\,\mathrm{m}$ (Sündermann and Pohlmann, 2011) connected to the North Atlantic via the English Channel in the south-west, and openly towards the north across the European continental shelf. The North Sea is exposed to tides whose tidal range is between approximately $1.8\,\mathrm{m}$ and $3.4\,\mathrm{m}$ in the southeastern German Bight. The main North Sea tidal wave is turning anti-clockwise

and drives the large scale current system (Sündermann and Pohlmann, 2011). In the southern North Sea, SPM and dissolved nutrients that entered though the river Rhine are transported eastward along the Dutch and German East-Frisian shore by this large scale current system. In addition, the German rivers Ems, Weser and Elbe discharge into the German Bight. As a result of the riverine nutrient input, the German Bight features a generally high primary productivity (Joint and Pomroy, 1993) while it possesses pronounced heterogeneity with strong cross-shore gradients in i) nutrients (Brockmann et al., 1990; Ebenhöh et al.,

2004), and ii) SPM concentration and composition (Eisma and Kalf, 1987). The near-coastal waters of the German Bight are characterized by relatively high SPM concentrations of few to several hundred gram per cubic meter in near-bottom regions. SPM concentrations possess a pronounced seasonality showing higher values in winter than in summer. This seasonal pattern is reflected in the shallow parts of the tidal back barriers of the Wadden Sea by the bottom fraction of fine, cohesive sediments accumulating during spring and summer due to aggregation and subsequent increased sedimentation (Chang et al., 2006). The

sediment catchment area for the Dutch and East-Frisian Wadden Sea system, an area sheltered from the North Sea by a chain of islands, is hypothesized to be defined by a *line-of-no-return* located alongshore off the coast (Postma, 1984).



## 2.2 Sampling and processing of observational data

Measurements were carried out as part of the 'Coastal Observing System for Northern and Arctic Seas' (COSYNA: www.cosyna.de; see also Baschek et al., 2016). Sensors were mounted on-board a towed vehicle (ScanFish, Mark II, EIVA a/s, Denmark). The ScanFish was remotely operated behind a vessel sailing with a speed of 6 to 8 knots. Cruises were carried out during May, July/August, September (2009), March, May, July, September (2010) and April, June and September (2011) during moderate weather conditions. The transect grid covered the German Exclusive Economic Zone (Fig. 1). The ScanFish was forced to operate in nearly V-curved undulating path mode between approximately $3\,\mathrm{m}$ distance to water surface and sea bottom with a vertical speed of $0.4\,\mathrm{m\,s^{-1}}$ and a sampling rate of $11\,\mathrm{s^{-1}}$, yielding a vertical data spacing of about $0.04\,\mathrm{m}$. Our analysis considered measurements of specific conductivity (Conductivity Sensor, ADM Elektronik, Germany), water temperature $T$ (PT100, ADM Elektronik, Germany), pressure $p$ (PA-7, Keller AG, Switherland), optical turbidity (Seapoint Turbidity Meter $880\,\mathrm{nm}$, Seapoint Sensors inc., USA), and chlorophyll $a$ fluorescence F (TriOS MicroFlu-chl, TRIOS Inc., Germany). Specific conductivity and $T$ were calibrated against reference standards directly before and after the ship surveys. Potential water density $\rho_\theta$, expressed as $\sigma_T = \rho_\theta - 1000\,\mathrm{kg\,m^{-3}}$, was calculated according to the EOS-80 equations of state (Fofonoff and Millard, 1983). Thermal lag effects were visible in $\sigma_T$ around thermoclines, but neglected since the selection criteria for profiles as described below attenuated their relevance for this analysis. During post-processing, data generally underwent tests for any stuck values and spikes, and were finally visually inspected to remove remaining obvious faults. Turbidity, measured in formazine turbidity unit (FTU), was converted to SPMC using a factor of $1.08\,\mathrm{g\,(dry\text{-}weight)\,m^{-3}\,FTU^{-1}}$ determined by linear regression (r$^2$=0.967; Röttgers et al., 2011). Laboratory investigations revealed a sensitivity of the fluorescence signal to turbidity. Therefore, a correction factor was applied to measured fluorescence. The factor of $(0.32 + (1 - 0.32)\exp(-0.025\,\mathrm{turbidity}))^{-1}$ was experimentally determined with commercially available chlorophyll $a$ dissolved in varying formazine concentrations. As a final step, data sets were split into separate up- and down casts between consecutive vertical extrema of the undulating flight path. Casts with deficient pressure data were discarded.

## 2.3 Data processing and sinking velocity extraction

Sinking velocities were obtained by fitting an analytical solution for the vertical distribution of SPMC to observations. Assuming steady state and neglecting horizontal advection, the SPM transport equation in time $t$ for concentration $C$ reduces to

$$\frac{\partial}{\partial t}C = \frac{\partial}{\partial z}k_\mathrm{v}\frac{\partial}{\partial z}C - \frac{\partial}{\partial z}(w_s C) = 0 \tag{1}$$

and describes the balance between fluxes in the vertical direction $z$ due to sinking and turbulent eddy diffusivity $k_\mathrm{v}$. If we assume that the sinking time scale is larger than the tidal period, we can simplify Eq. (1) by using the average $k_v$ of a profile. If we further allow fluxes across the profiles borders, we can derive an analytical model for depth-dependent SPMC

$$C_\mathrm{m}(z) = \frac{\lambda \exp(\lambda z^*/H_\mathrm{p})}{\exp(\lambda)-1}\langle C\rangle \quad \text{with } \lambda = \frac{w_s H_\mathrm{p}}{\langle k_v\rangle} \tag{2}$$



where $z^* = z - z_1$ is the actual minus the upper depth $z_1$, where a profile of height $H_\mathrm{p}$ starts. Application of the analytical solution to observed SPMC profiles required information on correspondent $k_\mathrm{v}$. These, and additionally energy dissipation rate $\epsilon$ and water density $\sigma_T$ were obtained from hydrodynamical simulations of Gräwe et al. (2015), based on a 1 nautical mile numerical model of the North Sea and the Baltic Sea. In the vertical, 42 terrain-following levels were used. The hydrodynamic
parameters were stored as 2 hourly snapshots.

Hydrodynamic model results for $k_v$, $\epsilon$, and $\sigma_T$ were linearly interpolated in space and time to extract a corresponding value for every measured data point. Even though state-of-the-art hydrodynamical models perform generally very well, they may locally exhibit discrepancies to observations. To eventually discriminate for lacking congruency, both, observed and modeled $\sigma_T$, were interpolated on a common vertical grid of $\Delta z = 0.05\,\mathrm{m}$ and filtered by applying a least-square straight line fit to
the data and a 'natural' cubic spline interpolant. The latter had a weight of 80 % to produce a smooth interpolation curve $\sigma_T'(z')$. Subtracting the respective vertical mean resulted in profiles $\widetilde{\sigma_T}$ for both observed and modeled data. We constrained our further analysis to casts satisfying two criteria. First, after subtraction of modeled $\widetilde{\sigma_T}$ from observed data, the standard deviation should not exceed $\mathrm{std}(\Delta\widetilde{\sigma_T}) \leq 0.015\,\mathrm{kg\,m^{-3}}$. Second, the density gradients defined as $\delta_z\sigma_T = (\langle\sigma_T'(\text{last meter})\rangle - \langle\sigma_T'(\text{first meter})\rangle)/H$, where $\langle x\rangle$ generally represents the ensemble mean of a variable $x$, should not exceed a discrepancy
of $\Delta\delta_z\sigma_T \leq 0.015\,\mathrm{kg\,m^{-3}\,m^{-1}}$ between observed and modeled data. Both limits for $\mathrm{std}(\Delta\widetilde{\sigma_T})$ and $\Delta\delta_z\sigma_T$ were somewhat arbitrary and therefore considered in a Monte-Carlo-type simulation (see below).

ScanFish cruises covered well-mixed coastal, but partly also stratified waters in the inner German Bight. A consistent approach to retrieve $w_s$ was required to meet both conditions. We chose to split casts into subprofiles, if needed, and to fit the single analytical solution, Eq. (2), to (sub-) profiles. If observed $\delta_z\sigma_T < 5 \cdot 10^{-4}\,\mathrm{kg\,m^{-3}\,m^{-1}}$, the whole profile was used for
fitting, otherwise the cast was split. The co-occurrence of strong gradients in $\sigma_T$ and SPMC can indicate dampening of turbulent mixing and potential particle properties' changes. Thus, end points of subprofiles were set, where regions in the profile with

$$\left|\frac{1}{C'}\frac{\partial C'}{\partial z'} \cdot \frac{1}{\sigma_T'}\frac{\partial \sigma_T'}{\partial z}\right| > \left\langle\left|\frac{1}{C'}\frac{\partial C'}{\partial z'}\right|\right\rangle\left\langle\left|\frac{1}{\sigma_T'}\frac{\partial \sigma_T'}{\partial z}\right|\right\rangle \tag{3}$$

occur and reach their maximum. $C'$ is the SPMC smoothing spline-interpolated analogously to $\sigma_T$. Start points of the subpro-
files were either the first data point from surface or where regions defined by Eq. (3) end with increasing depth. Only subprofiles longer than $4\,\mathrm{m}$ were considered in the further analysis.

The analytical model, Eq. (2), was fitted to the original observation of each (sub-) profile with $N_p$ data points, and the cost function

$$\mathrm{err} = \frac{1}{N_p}\sum_{i=1}^{N_c}\frac{(C(z_i) - C_\mathrm{m}(z_i))^2}{0.5\,(\delta_R C^2 + \delta_P C^2)} \tag{4}$$

was calculated accordingly. $\delta_P C^2$ and $\delta_R C^2$ represent the variance of concentration of a profile and in a region, respectively. The latter is defined as the variance for measurements around the profile within $\pm 5$ min. to account for higher variabilities in coastal than in open water regions found in a pre-analysis of the data. Only profiles below a cost function value of 0.05 were considered for the analysis. The variables SPMC, F/SPMC ratio and $w_s$ of these profiles were binned according to modeled $\epsilon$





and eventually averaged bin-wise, resulting in the respective bin-wise ensemble means $\langle \mathrm{SPMC} \rangle$, $\langle \mathrm{F/SPMC} \rangle$, $\langle w_s \rangle$, and $\langle \epsilon \rangle$. To test for significant changes of binned $\langle w_s \rangle$ with $\langle \epsilon \rangle$, we applied a Mann-Whitney U test with a significance level of $p < 0.05$ for each binned $\langle w_s \rangle$ against each other. In summary, approximately $67\,\%$ of initially measured $\approx 68.000$ profiles passed the congruency check with modeled ones. After application of the cost function threshold, this resulted in $\approx 12.260\ w_s$-values.

5    All applied threshold values were carefully selected by visual inspection during each filtering step. To quantify their influence on the result, we performed a Monte-Carlo-type simulation with a variation by $\pm 50\,\%$ for the following parameters: $\mathrm{std}(\Delta \widetilde{\sigma_T})$, $\Delta \delta_z \sigma_T$, $\delta_z \sigma_T$, and the cost function error. The analysis was repeated for variations of all parameters against each other. For each parameter variation, binned mean values for $\langle w_s \rangle$ versus $\epsilon$ were calculated, eventually averaged and their standard deviation $\sigma$ quantified (see Fig. 3 C).

## 2.4    Conceptual cross-coastal sinking velocity model

We introduce and apply a conceptual model for the cross-shore variation of $w_s$ as a tool to interpret our results. According to Stokes (1851), $w_s$ of a particle of diameter $D$ can be described by

$$w_{\mathrm{s}} = \frac{1}{18\,\mu} \left( \rho_{\mathrm{f}} - \rho \right) g\, D^2 \quad , \tag{5}$$

where $\mu$ is the dynamic viscosity, $\rho$ is the water density, and $g$ the gravitational acceleration constant. The floc density $\rho_{\mathrm{f}}$ is strongly related to the particles' composition and structure. The latter can be described operationally as fractal dimension-like ($d_{\mathrm{f}}$) according to e. g. Kranenburg (1994), who derived the excess floc density $\Delta \rho_{\mathrm{f}} = \rho_{\mathrm{f}} - \rho$ for a particle composed of primary particles of diameter $D_{\mathrm{p}}$ and density $\rho_{\mathrm{p}}$:

$$\Delta \rho_{\mathrm{f}} = \left( \rho_{\mathrm{p}} - \rho \right) \left( \frac{D_{\mathrm{p}}}{D} \right)^{3 - d_{\mathrm{f}}} \quad . \tag{6}$$

Following the approach of Kranenburg (1994), Maggi (2009) derived the excess floc density for an aggregate composed of mineral and organic particles with density $\rho_{\mathrm{s}}$ and $\rho_{\mathrm{o}}$, respectively,

$$\Delta \rho_{\mathrm{f}} = \left( \omega \Delta \rho_{\mathrm{s}} + (1 - \omega) \Delta \rho_o \right) \cdot \left( \frac{D_{\mathrm{p}}}{D} \right)^{3 - d_{\mathrm{f}}} \quad , \tag{7}$$

under the assumption of equally sized primary particles of diameter $D_{\mathrm{p}}$. The relation of primary particles number of mineral and organic origin, $n_{\mathrm{s}}$ and $n_{\mathrm{o}}$ respectively, is expressed by $\omega = n_{\mathrm{s}}/(n_{\mathrm{s}} + n_{\mathrm{o}})$. For $w_s$ of an aggregate, it follows

$$w_{\mathrm{s}} = \frac{1}{18\,\mu} (\omega \Delta \rho_{\mathrm{s}} + (1 - \omega) \Delta \rho_o)\, g\, D_{\mathrm{p}}^{3 - d_{\mathrm{f}}} D^{d_{\mathrm{f}} - 1} \quad . \tag{8}$$

25    This simple $w_s$ model, Eq. (8), allows for calculation of trends in $w_s$ on a cross-shore transect. Based on observations, we presumed the general following parameter changes from coastal to open waters: *i)* average primary particle size and aggregate size increase, *ii)* particles become more organic, and *iii)* more fluffy, thus $d_f$ decreases. We assumed a linear de- or increase of parameters with distance from coast. As boundary conditions for coastal waters, we therefore assume an average $\langle D_{\mathrm{p}} \rangle = 4\,\mu\mathrm{m}$ (mainly cohesive sediment dominated; Winterwerp, 1998), $\langle D \rangle = 100\,\mu\mathrm{m}$ (e. g. Fettweis et al., 2006), $\omega = 0.9$ (resembling a





loss on ignition (LoI) of $\approx 0.04$ according to Fettweis, 2008), and $d_f = 2.0$ as an average coastal fractal dimension (Winterwerp, 1998). By contrast, for open waters we presume $\langle D_p \rangle = 10\,\mu\text{m}$ for algae-dominated particles, $\langle D \rangle = 500\,\mu\text{m}$ (assuming similar trends as in van der Lee et al., 2009, for the Irish Sea), $\omega = 0.05$ (LoI$\approx 0.89$ within the range described by Eisma and Kalf, 1987), and $d_f = 1.6$ as a typical value for fluffy aggregates (Logan and Alldredge, 1989; Li and Logan, 1995). The densities of primary particles are set to $\rho_p = 2650\,\text{kg m}^{-3}$ (according to Maggi, 2009) and $\rho_o = 1100\,\text{kg m}^{-3}$ (as an average value of the range given in Fettweis, 2008). The water density and the dynamic viscosity are set to $\rho = 1000\,\text{kg m}^{-3}$ and $\mu = 0.001\,\text{kg m}^{-1}\,\text{s}^{-1}$, respectively. In order to demonstrate the parameter's impact on $w_s$, each parameter is varied separately for an assumed particle of diameter $D = 100\,\mu\text{m}$, $\omega = 0.5$, $D_p = 4\,\mu\text{m}$, and $d_f = 2$.

## 3 Results

### 3.1 Cross-coastal gradients

Energy dissipation rates $\epsilon$ generally possess spatio-temporal variability. Most relevant for this study, vertically and temporally averaged model-derived $\epsilon$ exhibit a cross-shore gradient from the inner German Bight towards the coast. Values range between $\epsilon \approx 10^{-9}\,\text{m}^2\,\text{s}^{-3}$ and $\epsilon \approx 10^{-4}\,\text{m}^2\,\text{s}^{-3}$, respectively (Fig. 2).

Along this range, mean SPMC first increased from values below $1\,\text{g m}^{-3}$ to approximately $10\,\text{g m}^{-3}$. Above $\log_{10}(\epsilon) = -5.5$, however, mean SPMC moderately decreased to approximately $6\,\text{g m}^{-3}$ and increased again to $10\,\text{g m}^{-3}$ with further increasing $\epsilon$ (Fig. 3 A) towards the coast.

The potential significance of algae and their products for particle composition is depicted by the mean ratio between measured fluorescence and SPMC. This ratio shows rather high variability for $\log_{10}(\epsilon) < -6$ (Fig. 3 B). With further increasing $\epsilon$, the ratio drops and levels at approximately a fourth of the open German Bight value for $\log_{10}(\epsilon) > -5$.

Average sinking velocities $\langle w_s \rangle$ show very low values of order $10^{-6}\,\text{m s}^{-1}$ to $10^{-5}\,\text{m s}^{-1}$ in regions of $\log_{10}(\epsilon) < -7.5$ (Fig. 3 C). At higher $\epsilon$, $\langle w_s \rangle$ increased significantly reaching a maximum of about $7 \cdot 10^{-4}\,\text{m s}^{-1}$ around $\log_{10}(\epsilon) \approx -5.5$. This maximum $\langle w_s \rangle$ around $\log_{10}(\epsilon) \approx -5.5$ coincided with the increase of SPMC, and the drop in fluorescence to SPMC ratio. With further increasing $\epsilon$, $\langle w_s \rangle$ decreased again to values of $\langle w_s \rangle \approx 3 \cdot 10^{-4}\,\text{m s}^{-1}$. The Monte-Carlo-type simulation to estimate parameter uncertainties exhibits the same pattern, expressed as $\sigma$ and $2\sigma$ confidence levels, 68 % and 95 % respectively (Fig. 3C), around their ensemble mean, and thus underpins the results of $\langle w_s \rangle$ along $\epsilon$.

### 3.2 Conceptual model

Settling velocity of an average particle (as parametrized in Sec. 2.4) changes, after variation in single parameters along the transect from open to coastal waters results, in an almost linear way (Fig. 5, upper four panels). Flocs would sink less rapid because of decreasing diameter of the aggregate or the primary particles in the coastal region. By contrast, increasing amount of sediment particles and fractal dimension would both lead to increased $w_s$. As an overall effect, when considering all parameters, $w_s$ reaches a maximum in between the near-shore and open waters (lowest panel of Fig. 5).



## 4 Discussion

### 4.1 Sinking velocity on a gradient of prevailing energy dissipation rate

Sinking velocities have been determined along a cross-shore transect in the German Bight defined by the prevailing $\epsilon$. For the reconstruction of $\langle w_s \rangle$, we had to assume congruency between *in situ* measurements and hydrodynamic model results within a range defined by the applied filters. The estimated $\langle w_s \rangle$ of order $10^{-6}\,\mathrm{m\,s^{-1}}$ to $5 \cdot 10^{-4}\,\mathrm{m\,s^{-1}}$ in low and high energy dissipation regions, respectively (Fig. 3), compare very well with former *in situ* studies in the German Bight and adjacent areas. Puls et al. (1995) found $w_s \approx 10^{-6}\,\mathrm{m\,s^{-1}}$ to $2 \cdot 10^{-5}\,\mathrm{m\,s^{-1}}$ for the open German Bight, a low $\epsilon$ region, and Pejrup and Mikkelsen (2010) reported median $w_s$ of $1 \cdot 10^{-4}\,\mathrm{m\,s^{-1}}$ to $11 \cdot 10^{-4}\,\mathrm{m\,s^{-1}}$ for the Rømø and Høyer Dyb, Danish Wadden Sea where $\epsilon$ is comparably high. Our estimated $\langle w_s \rangle$ are also well in the range found in other regions, e. g. in Chesapeake Bay (Gibbs, 1985) or at the Belgian Coast (Fettweis and Baeye, 2015). In agreement with former studies, obtained $\langle w_s \rangle$ increase with increasing $\langle SPMC \rangle$ (Fig. 4), which gave further confidence in the methodological approach. The correlation is, however, rather poor compared to former local studies and can be explained by the heterogeneity of the German Bight system, seasonal effects and the intrinsically high variability of sinking velocities (Fettweis, 2008). With respect to prevailing $\epsilon$, the significant maximum of $\langle w_s \rangle$ at $\langle \epsilon \rangle \approx 10^{-5.5}\,\mathrm{m^2\,s^{-3}}$ can be explained by the balancing between aggregation and fragmentation. Both processes are controlled by turbulent shear, generated by energy dissipation, SPM volume concentration, and adhesion properties of particles involved. Previous flocculation modeling studies pointed to a formation of a $w_s$ maximum along $\epsilon$ (Winterwerp, 1998; Pejrup and Mikkelsen, 2010). Our findings of a maximum in $\langle w_s \rangle$ between $\epsilon \approx 10^{-6}\,\mathrm{m^2\,s^{-3}}$ and $10^{-5}\,\mathrm{m^2\,s^{-3}}$ thus agree well with, and support, former theoretical studies. Hence, as already suggested by Dyer (1989) and Pejrup and Mikkelsen (2010), in addition to SPMC, turbulent shear can be regarded as the major determinant for $w_s$.

### 4.2 Sinking velocity as a result of SPM concentration, composition and a turbulence gradient

Our study additionally suggests a change in particle composition with $\epsilon$ as proxied by the fluorescence to SPMC ratio. With $\epsilon$, SPMC increases towards the coast and mineral fraction becomes dominant compared to organic particles at low $\epsilon$ (Eisma and Kalf, 1987). Hence, density and likely other physico-chemical properties of primary particles such as size and adhesion change accordingly and thus influence $w_s$ along a cross-shore transect. Our conceptual model illustrates the effect of varying particle properties on $w_s$ independent of the observations used in our data analysis. However, the course of the parameters implicitly presumes certain environmental conditions, as e. g. size of particles is, among others, a function of SPMC and shear. Single parameter changes in the conceptual model resulted in mostly linear responses in $w_s$, while considering all parameters led to a maximum in $w_s$ on the conceptual cross-shore transect, in alignment with our findings in the data analysis. While the cross-shore distribution of $w_s$ is predominantly controlled by prevailing shear, the conceptual model indicates an additional relevance of physico-chemical properties for the formation of the high sinking velocity zone.



### 4.3 Implications of a cross-coastal maximum of sinking velocity for biogeochemical cycling in the coastal zone

The region of $\log_{10}(\epsilon) \approx -5.5$ with highest $\langle w_s \rangle$ coincides with a zone located off the coast at depth of about $15\,\mathrm{m}$ to $20\,\mathrm{m}$. It is accompanied by a strong gradient in SPM concentration and likely composition as depicted by the F/SPMC ratio. This suggest that the region can be considered as a transition zone, hindering mineral particles to escape further off-shore. Such trapping

has been previously described conceptually for other regions, e. g. by Mari et al. (2012) and Ayukai and Wolanski (1997) for river plumes. High $\langle w_s \rangle$ in the transition zone implies an enhanced link between pelagic and near-bottom processes. Among them, different transport mechanisms have been discussed that potentially lead, on average, to a net transport of fine sediments shore-wards into the tidal back barriers. *Settling lag* is caused by the different time durations that it takes in different water depths for particles to sediment after the bottom shear stress for erosion falls below the critical one (Postma, 1961). Another

process is *scour lag* that potentially arises from lower bottom shear stresses needed to keep particles in suspension than for their erosion in combination with an asymmetry in the flow (Van Straaten and Kuenen, 1957). These effects were investigated under different topographical settings (van Maren and Winterwerp, 2013) and the authors concluded that $w_s$ in the range of $0.5\,\mathrm{mm\,s^{-1}}$ and $1\,\mathrm{mm\,s^{-1}}$ led to the highest deposition rates, and underlined the necessity of fine mineral particle flocculation for the build up of tidal flat systems. Recently, another potential major driver, the estuarine circulation, has been suggested

for sediment accumulation in the Wadden Sea. Residual currents of the estuarine circulation are driven by density gradients that can be caused by i) horizontal temperature gradients, ii) differential precipitation, and iii) river discharges, and iv) all possible superpositions. Residual currents of the estuarine circulation point on-shore in near-bottom, and off-shore in surface waters (Burchard et al., 2008; Flöser et al., 2011). This probably causes a net SPM flux into the Wadden Sea (Burchard et al., 2008, 2013). The latter is hypothesized to act as bio-reactor, where organic components of SPM become re-mineralized and are

exported in dissolved form (Postma, 1984; Ebenhöh et al., 2004; Grunwald et al., 2010) by the off-shore-directed component of the estuarine circulation. Along this pathway, dissolved nutrients are assimilated by phytoplankton and thus transferred to bio-particulates. This likely happen especially within the region of prevailing $\epsilon \approx 10^{-5.5}\,\mathrm{m^2\,s^{-3}}$ where phytoplankton experience favorable growth conditions as found by Pingree et al. (1978) for different places on the North-European continental shelf. Algae exudate extracellular polymeric substances (EPS) like TEP that can undergo and mediate flocculation by bridging and

embedding minerals. The latter ballasts the organic matrix leading to enhanced sinking velocities, as formerly described for the deep ocean (Passow, 2004; De La Rocha and Passow, 2007), which again link pelagic processes to the residual near-bottom currents of the estuarine circulation. Summarized, this would imply that the transition zone characterized by its high $\langle w_s \rangle$ accompanied with strong SPMC gradients acts as an important off-coastal feature for closing the near-shore nutrient and mineral cycle. Such a biologically mediated trapping mechanism would retain nutrients and minerals in the coastal zone and

inside the Wadden Sea back barriers.

### 4.4 Spatial biogeochemical implications of a coastal transition zone

Typically, spatial information on biochemical variables are needed to better understand system-wide cycling and fate of matter. Unfortunately, averaging of $w_s$ over bins of eps means in principle loss of direct spatial reference. However, the elaborations





made it possible to identify a general pattern of $\langle w_s \rangle$ with a defined maximum along prevailing energy dissipation rates. Even though we here neither resolve tidal cycles nor consider potential seasonal variation of $\langle w_s \rangle$ along $\langle \epsilon \rangle$, a map of $\langle w_s \rangle$ allows for an insight into spatial distribution and variability of $\langle w_s \rangle$. It was re-constructed by mapping the found bin-wise relationship between $\langle \epsilon \rangle$ and $\langle w_s \rangle$ (Fig. 3C) onto the respective spatial $\langle \epsilon \rangle$ (Fig. 2). It must be highlighted that local, short term

*in situ* measurements would most likely deviate from the obtained averaged picture (Fig. 6) that would be challenging to derive by *in situ* measurements. Nevertheless, strong spatial variability is visible with a pronounced cross-coastal gradient of $\langle w_s \rangle$ featuring a maximum of sinking velocities along the coastline as indicated before. However, this maximum varies locally and is particularly less pronounced at the northern German and southern Danish coast, where $\langle w_s \rangle$ declines to values that are rather in the range of $\langle w_s \rangle$ in deeper waters in the southern German Bight. Applying the aforementioned concept of the transition zone as

an off-coastal closing mechanism for nutrient cycling to the two contrasting German Wadden sea regions, East Frisian Wadden Sea and North Frisian Wadden Sea, in the latter particularly the Sylt-Rømø Bight, may help to better understand regional differences of nutrient concentrations. Lower nutrient concentrations and thus lower eutrophication levels in the Sylt-Rømø bight compared to the East Frisian Wadden sea are regularly observed (van Beusekom et al., 2009). In this case, the maximal $\langle w_s \rangle$ calculated in front of the Sylt-Rømø tidal inlet, amounts only to $4 \cdot 10^{-4}\,\mathrm{m\,s^{-1}}$, about half the magnitude found in front

of the other Wadden Sea regions. Hence, the ability for nutrient retention is diminished and would lead to generally lower nutrient concentrations in similarly affected Wadden Sea regions. By contrast, the potentially pronounced retention capacity of the transition zone, as predicted for the Dutch coast and German East-Frisian islands may contribute to the phenomenologically described *line of no return* (Postma, 1984), expected further off the coast and characterized by low SPM concentration. The spatial distribution of $\langle w_s \rangle$ is probably also reflected in the SPM concentration and their cross-coastal gradients (Fig. 7). On

average, lower $\epsilon$ accompanied by lower SPM concentrations and smaller horizontal gradients are found along the North-Frisian than the Dutch and East-Frisian coast (Fig. 2 and 7). In combination, these conditions in particular along the Sylt-Rømø islands suppress the establishing of a defined transition zone. Accordingly, lower $w_s$ should be found as also implied by Fig. 6. As a result, retention efficiency is likely reduced which leads to the observed lower nutrients concentrations compared to the East-Frisian Wadden Sea (van Beusekom et al., 2009). Since the driving mechanisms behind the transition zone are probably

ubiquitous for coastal marine systems with sufficiently strong cross-shore water density gradients generated by freshwater run-off, precipitation-evaporation balance or heat fluxes, local hydrodynamics determine its eventual formation. This implies that the concept of the transition zone with its retention capacity for nutrients may be applicable to other coastal marine and estuarine systems in general, and will help to better understand different nutrient cycling behavior among those systems.

### 4.5 Biological modulation of the transition zone?

There is increasing evidence that sticky algae excretions such as transparent exopolymer particles mediate aggregation and increase shear resistance of particles against fragmentation, potentially enabling algae to clear the water column from mineral load (Fettweis et al., 2014). As shown in the conceptual model (Sec. 2.4), high $w_s$ occur due to the interplay of flocculation and primary particle density and size. Similarly, Hamm (2002) found a non-linear effect of mineral-organic weight/weight-content on sinking velocity showing first an increasing $w_s$ for increasing mineral load, but a stagnation or even converse



effect when exceeding a certain threshold. To hypothesize, a high concentration of SPM with an optimal ratio between dense mineral and sticky bio-particles might exist in the transition zone under probably favorable turbulent conditions to form larger flocs compared to near-coast regions, and denser flocs compared to open waters. In sum this could lead to the found high $\langle w_s \rangle$. This rises the important question to what extent algae affect the transition zone and its seasonal location to sustain the

effective coastal nutrient cycle that is driven by the estuarine circulation. By sustaining the nutrient cycle, algae thus potentially support the development of the pronounced nutrient gradients towards the Wadden Sea. As a consequence, 3-D biogeochemical models require a representation of the tight coupling between sediment dynamics and biogeochemical, potentially even adaptive phytoplankton physiological processes to improve models' capability to estimate particulate matter fluxes.

## 5   Conclusions

The present study provides evidence for a maximum of sinking velocities along a cross-shore transect that is defined by a gradient of prevailing energy dissipation rates. Towards the off-shore, the enhanced sinking velocities are accompanied by a strong decline of SPM concentration, and an increasing F/SPM ratio. This suggests to define the region as coastal transition zone for SPM concentration and composition.

The transition zone is probably an important feature on the course from the coast to the continental shelf. It potentially acts

as a retention zone for dissolved nutrients leaking from near-shore waters by providing favorable conditions for algae growth. Phytoplankton take up nutrients and excrete extracellular polymeric substances that mediate flocculation processes. Embedding minerals into the organic matrix leads to enhanced sinking velocities. Algae thus seem to possess the ability to clear the water column, and link the off-shore dissolved nutrient fluxes to residual landward bottom fluxes. Consequently, these interlinked processes retain fine sediments and nutrients in coastal areas. The strength of the links eventually affects the eutrophication

state of the coastal region.

It is scarcely understood which relevance individual processes have in forming the transition zone. Algae with their seasonality seem to be strongly involved, thus further studies are needed to investigate the temporal and spatial extension of the transition zone. Long term or repeated spatial *in situ* measurements, including SPMC, $\epsilon$ profiles and particle properties such as size, $w_s$ and (volume-) composition, are required to underpin the indirectly found enhanced sinking velocities as feature of

the transition zone. Additionally, modeling approaches are required to gain a deeper process-based understanding, and to disentangle the closely linked processes. This also implies the necessity to incorporate biological-mineral interactions in models, especially, when system-wide studies are carried out.

*Author contributions.* R. Riethmüller planed and carried out the ScanFish measurements in the German Bight and took responsibility to ensure data quality. E. M. van der Lee conducted an intensive pre-analysis of the data. U. Gräwe provided hydrodynamical model results

from GETM for the times of the cruises. R. Hofmeister handled the 3D GETM results to transfer them to the ScanFish track. J. Maerz and K.W. Wirtz discussed and outlined the general research approach. J. Maerz carried out the data analyses and prepared the manuscript. All co-authors critically contributed to the discussion and to the interpretation of the results and the compilation of the manuscript.





*Acknowledgements.* The authors thank the crew of the research vessel RV Heincke, H. Rink, H. Thomas, M. Heineke, and R. Kopetzky for operating the ScanFish, for processing and quality assurance of the observational data and laboratory work, and L. Merckelbach and G. Flöser for fruitful discussions. This work was supported *i)* by the German Federal Ministry of Research and Education in the framework of the projects PACE (The future of the Wadden Sea sediment fluxes: still keeping pace with sea level rise?, FKZ 030634A) and MOSSCO (Modular System for Shelves and Coasts, FKZ 03F0667A), *ii)* by the Helmholtz society via the program PACES, and *iii)* by the Niedersächsisches Ministerium für Wissenschaft und Kultur (MWK) and the Niedersächsisches Ministerium für Umwelt und Klimaschutz (MUK) in the framework of the WiMo project (Scientific monitoring concepts for the German Bight). The financing of further developments of the IOW's Baltic Monitoring Program and adaptations of numerical models (STB-MODAT) by the federal state government of Mecklenburg-Vorpommern is greatly acknowledged by Ulf Gräwe. Supercomputing was provided by the North-German Supercomputing Alliance (HLRN). Observational and model data and related material are available for scientific purpose upon request to the corresponding author.



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





**Figure 1.** Map of water depth in the German Bight. Gray lines indicate the ScanFish sampling transects. The inset shows the North Sea and in red the drawn region. For East- and North-Frisian sub-regions see Fig. 6.





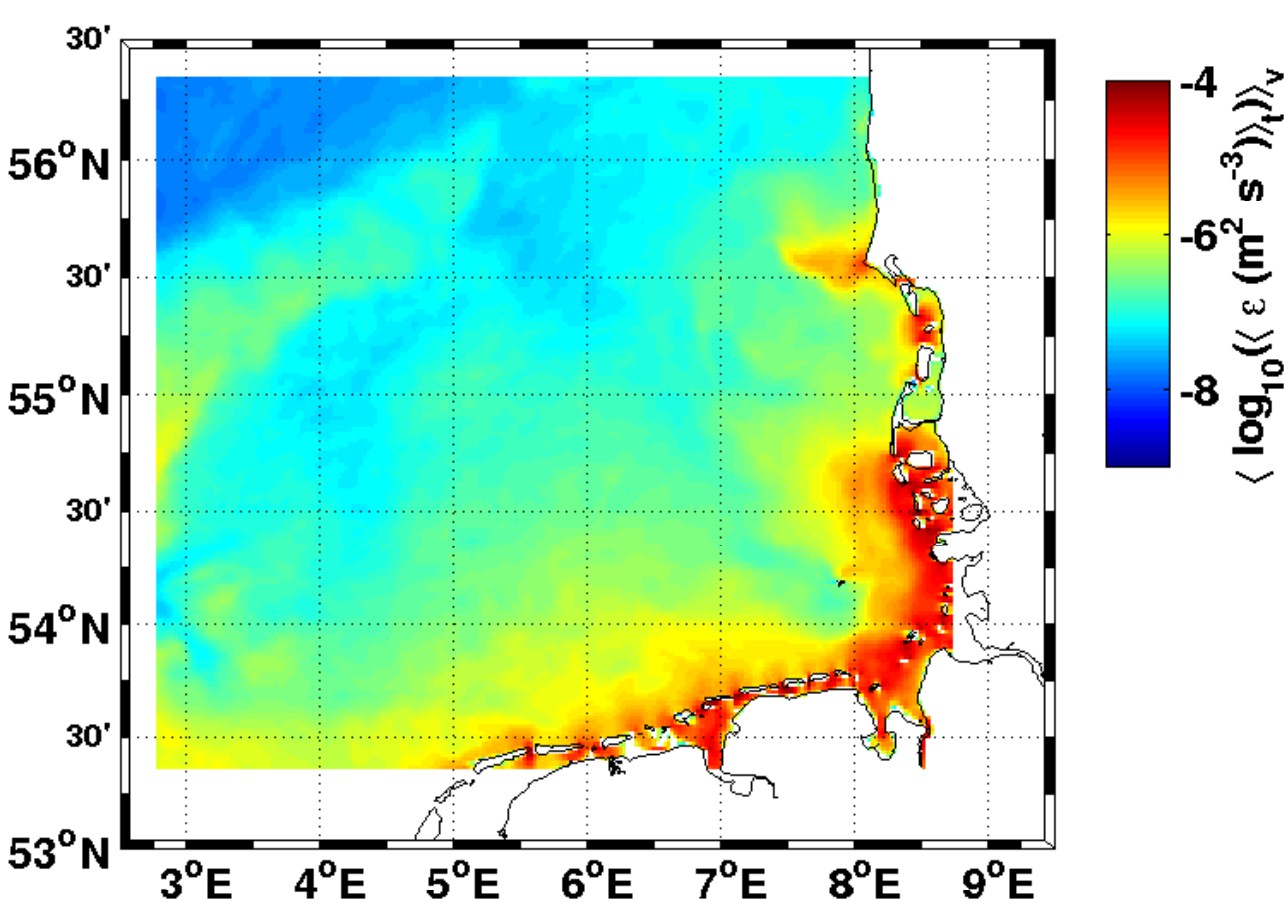

**Figure 2.** Map of GETM-calculated time- and depth-averaged energy dissipation rate $\epsilon$ for the times of the cruises.





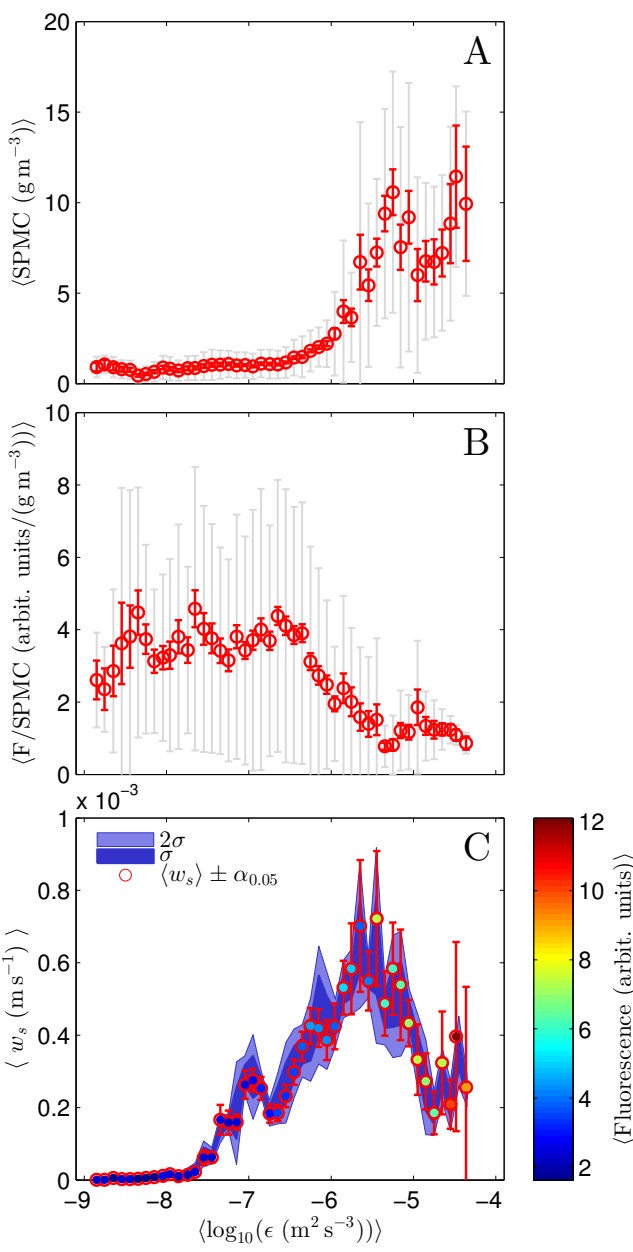

**Figure 3.** A: $\langle SPMC \rangle$, B: $\langle F/SPMC \rangle$ ratio and C: $\langle w_s \rangle$ versus $\langle \log_{10}(\epsilon(\mathrm{m^2\,s^{-3}})) \rangle$. Gray error bars represent the standard deviation and red error bars represent the confidence intervals ($\alpha$-quantile=0.05) for the averages in each bin. Average values for the bins were used for the color coding. In C: $\sigma$ and $2\sigma$ represent the confidence levels of $68\%$ and $95\%$, respectively, for the Monte-Carlo-type parameter variation.





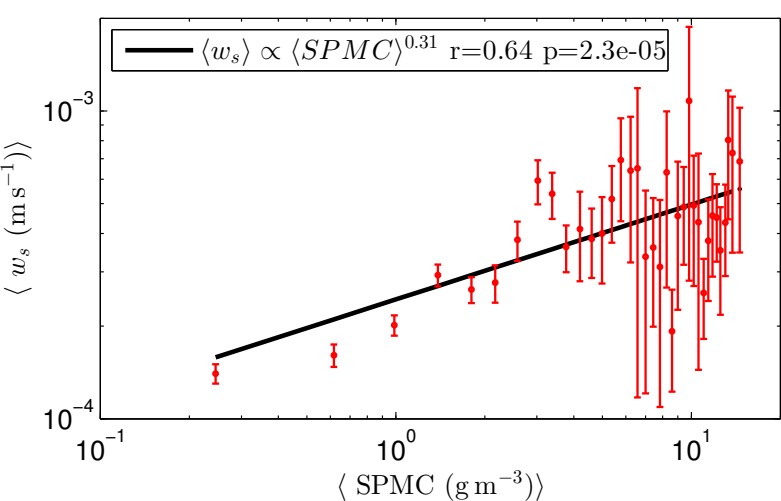

**Figure 4.** Scatterplot of $\langle SPMC \rangle$ and $\langle w_s \rangle$. Error bars represent the the confidence intervals ($\alpha$-quantile=0.05) for the averages in each bin. The rather poor correlation can be attributed to the heterogeneity of the German Bight system. See text for discussion.





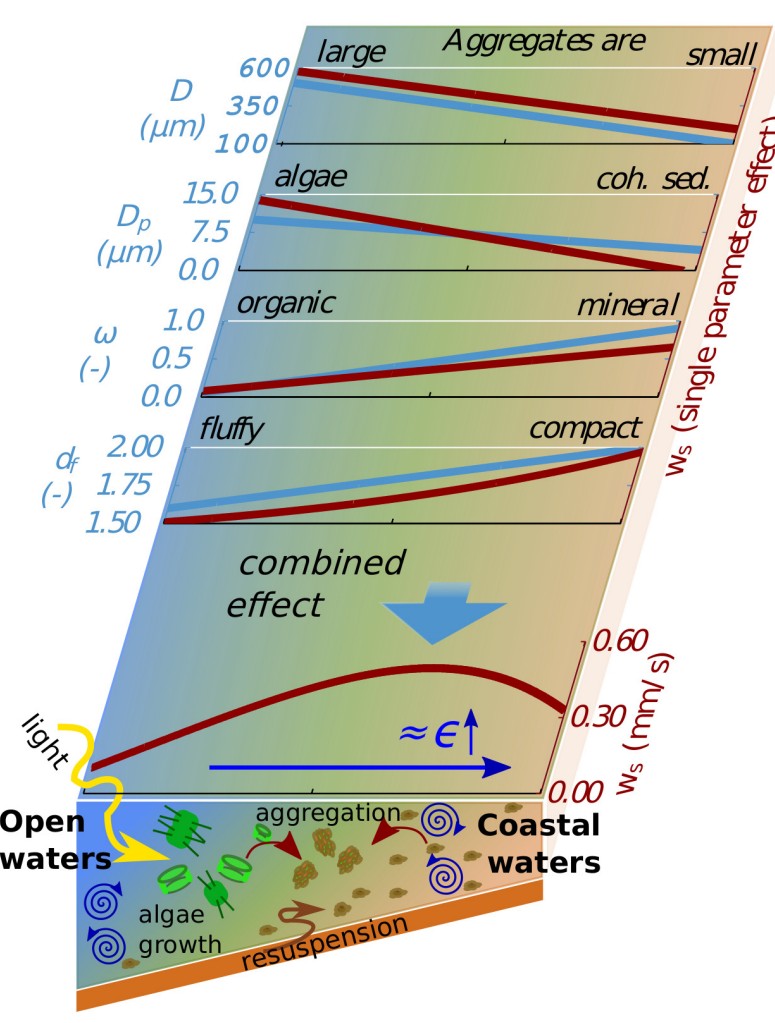

**Figure 5.** Schematic view of potential influences on sinking velocity from open waters (left) to coastal waters (right) and its total impact on the resulting sinking velocity (lowest panel) calculated according to Eq. (8). The sinking velocities shown in the upper 4 panels are based on a particle of $D = 100\,\mu\text{m}$, $\omega = 0.5$, $D_{\text{p}} = 4\,\mu\text{m}$, and $d_f = 2$ while the respective parameter shown in the panel is varied. (constant parameters are $\mu = 0.001\,\text{kg}\,\text{m}^{-1}\,\text{s}^{-1}$, $\rho = 1000\,\text{kg}\,\text{m}^{-3}$, $\rho_{\text{o}} = 1100\,\text{kg}\,\text{m}^{-3}$, and $\rho_{\text{s}} = 2650\,\text{kg}\,\text{m}^{-3}$).





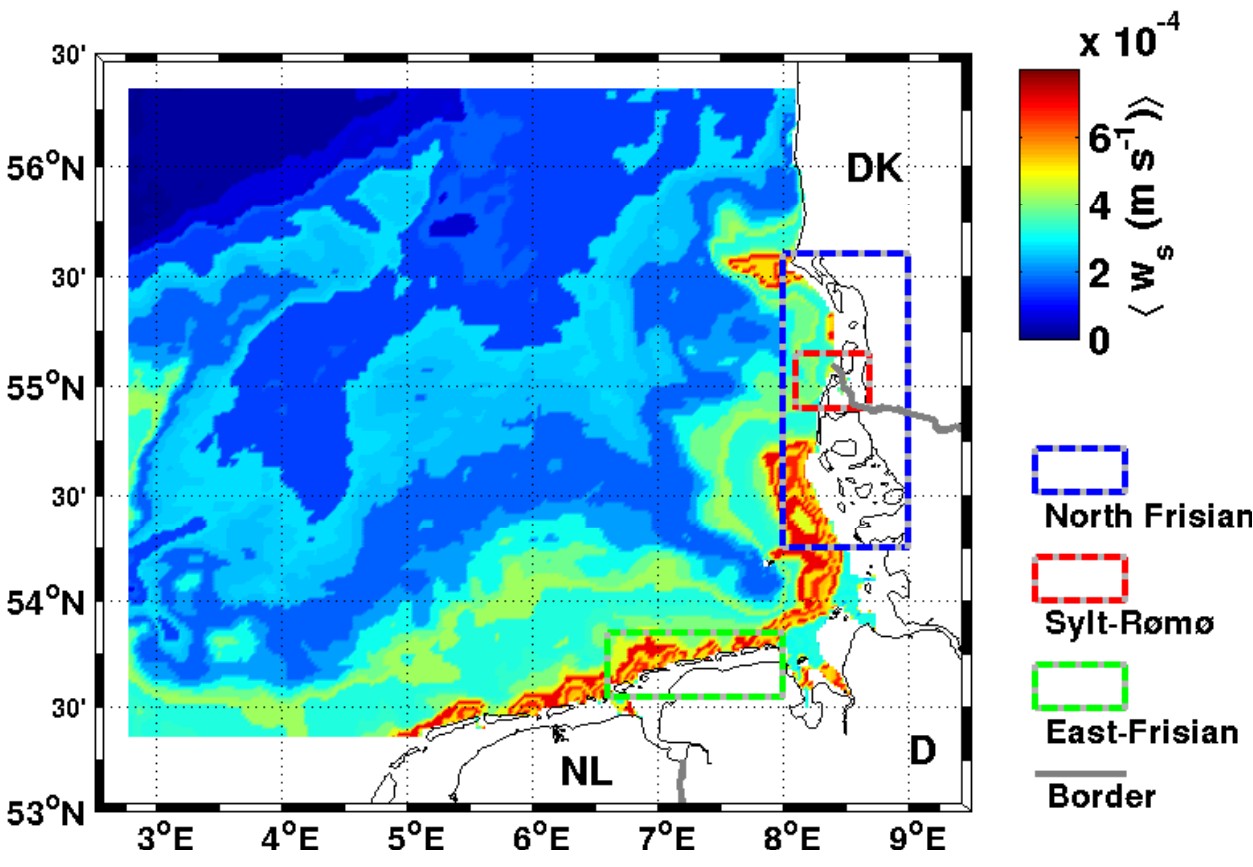

**Figure 6.** Map of spatial distribution of $\langle w_s \rangle$ in the German Bight. Results for $\langle w_s \rangle$ from Fig. 3 were mapped onto the modeled $\langle \epsilon \rangle$ shown in Fig. 2 to provide spatial information on potentially prevailing $\langle w_s \rangle$. Areas of water depth smaller than $5\,\mathrm{m}$ are not considered.





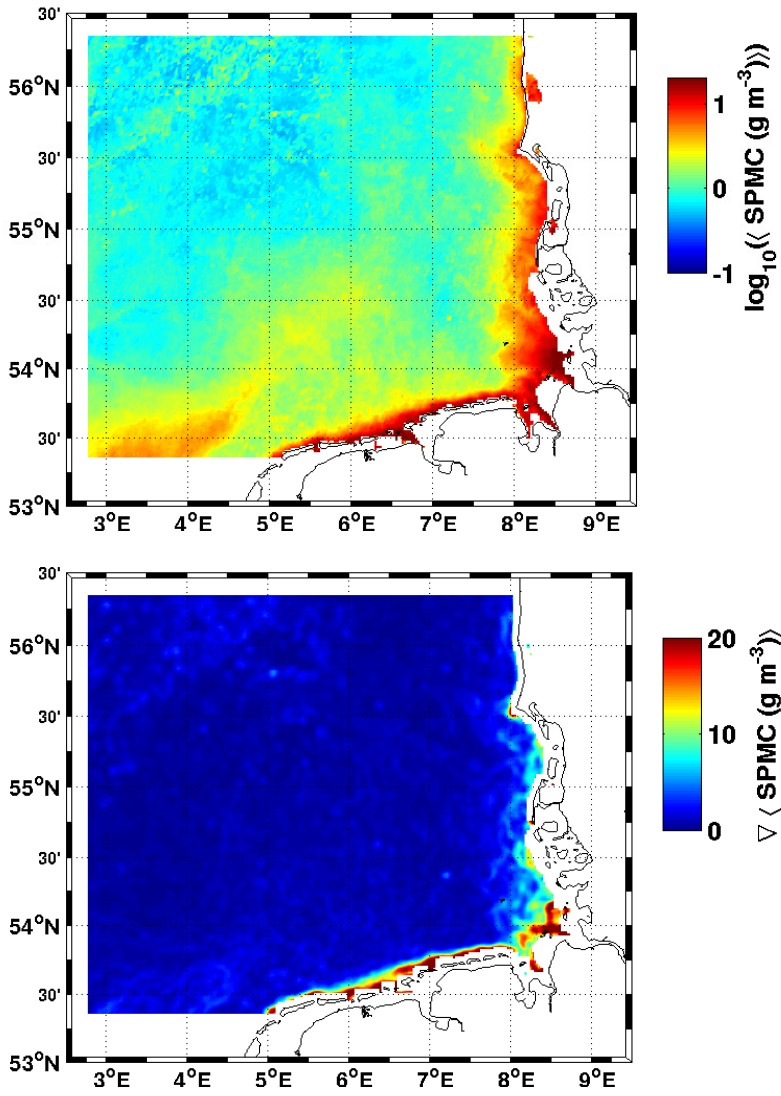

**Figure 7.** Top: MERIS-derived temporally-averaged SPM concentration in the German Bight. Averaging of available MERIS scenes was carried out for the times of the cruises. Regions of water depths smaller than 5 m were whited out. Bottom: Pixel-wise gradients of SPM concentration calculated by $((\partial_x \langle \mathrm{SPMC} \rangle)^2 + (\partial_y \langle \mathrm{SPMC} \rangle)^2)^{\frac{1}{2}}$, where $x$ and $y$ are east-west and south-north pixel directions, respectively. A Wiener filter of 3x3 pixel was subsequently applied. Notice the stronger gradients along the Dutch and East-Frisian coastline compared to the North-Frisian coast with the Sylt-Rømø-bight at 55 °N. For regions see also Fig. 6.