# Peer review of "Maximum sinking velocities of suspended particulate matter in a coastal transition zone"

_Biogeosciences, 2015_

## Referee Comment (RC1) · Anonymous Referee #2 · 7 Mar 2016

The manuscript aims to present new evidence to understand SPM dynamics in coastal seas, especially the mechanism that sustains the net-transport of SPM towards coastal systems, such as the Wadden Sea. The authors have used an approach that indirectly calculates the settling velocity in a cross-shore direction by using measured SPMC profiles and modelled turbulence data. The approach is original and the results are convincing. However, I am not in favor to call the obtained results an 'evidence', as the methods is based on a lot of assumptions that are not providing 'evidence based data'. Some of the assumptions used are further speculative and thus not supporting evidence. The latter would, e.g. be the case if settling velocity would have been measured directly. Nevertheless, even without direct evidence, the manuscript remains valuable

and will surely inspire other scientists to look for similar mechanism in other coastal regions or to adapt existing monitoring programs to collect more evidence based data to prove the hypothesis.

My major comment is therefore to not argue that you have found evidence to accept the hypothesis. Change the title accordingly and weaken the conclusions. As I have also some specific comments (see below) that are going further than minor changes I recommend to accept the manuscript only after major revisions.

Specific comments

SPM concentration or SPMC: try to be consistent. Sinking velocity or settling velocity: idem

p2 l27-28 "Cohesive sediments and POM can undergo aggregation and fragmentation processes that change transport properties and thus their sinking velocity". Change the sentence into: flocculation changes the settling velocity and thus the transport properties and not the opposite way. In previous sentence you make the difference between cohesive and non-cohesive minerals, thus between clays and the other minerals such as quartz or carbonates. Do you have evidence that only the cohesive minerals are involved in flocculation? What about very fine quartz or carbonates that can due to electric charges or the presence of specific organic molecules (EPS) be involved in flocculation?

p2 l37-39: "In shallow waters near the coast, where turbulence and thus resuspension are high, SPM concentration is usually enhanced and dominated by mineral particles with high densities". Better: The SPM concentration consist of flocs that are composed mainly of mineral particles. p2 l39-40: "By contrast, in deeper off-shore regions, SPM concentration is comparably low and consists to a higher extent of POM with low densities." Better: … SPM concentration is lower and the flocs are looser and more organic.

p2 l51: what do you mean by 'restructuring processes'?

p3 l67: add a reference for the 'typical power law relation'

p3 l94: an average depth of 80 m is not what I would call 'shallow'. Further, the water depth in the German Bight seems to be much less than 80 m from Fig 1.

p4 l99: The Rhine is not the only source of SPM and nutrient in the southern North Sea: see the general circulation pattern that brings Channel water into the North Sea and all the subsequent sources. What about the East Anglia plume that extends towards the German Bight?

p4 l110-113: I don't understand the hypothesis of 'line-of no-return'

p4 l119: The cruises are all in spring/summer period. Would the hypothesis of maximum settling velocity also be valid in winter? Do you have winter data?

p4 l123: sampling rate of 11 s-1: better 11 Hz

p4 l124: conductivity or specific conductance. I am not sure that the term 'specific conductivity' is used.

p4 l126: Is a Seapoint turbidity meter appropriate to be used in the high turbidity coastal areas where you say that SPM concentration is > a few 100mg/l.

p5 l147: "If we assume that the sinking time scale is larger than the tidal period". How valid is this assumption? The time scale of a tidal cycle is about 12.5 h (or if you consider ebb/flood: about 6h), and of the sinking time scale about 3-6h (w=1 mm/s in 10m water depth: 3h, ws=0.5 mm/s in 10 m water depth: 6h). This seems to me quiet similar. What are the consequences of this assumption on the results?

p5 l150: Cm is not defined. Is this the depth dependent SPMC?

p5 l160ff: How big is the difference between model results and observations. It would help a lot to better understand the procedure if you would show examples of the fitting,

interpolations etc.

p6 l174 On what is this criterion (< 0.0005 kg m-3 m-1) based see line 168 where it is 0.015.

p6 l175-176: "The co-occurrence of strong gradients in _T and SPMC can indicate dampening of turbulent mixing and potential particle properties' changes." Is it always the case that strong gradients in sigma T are co-occurring with those in SPMC? What if only one of the two parameters has strong gradients?

p6 line 189: On what is the cost function value of 0.05 based?

p6 line 190: the variable F is not defined. Why is it necessary to bin the data. If I understand well the methods than the data have already been interpolated on 5 cm resolution.

p7 l 209: "be described operationally as fractal dimension-like": what do you mean by 'operationally', skip '-like'

p8 l246-249: I don't agree on the potential significance depicted by F:SPMC ratio. Fig 3b shows in fact that F is increasing towards the high turbulent areas (towards the coast thus).

As SPMC is however stronger increasing than F the ratio is dropping. The highest alga concentrations are generally found in the nutrient rich, high turbid coastal areas.

Fig 4 is not in the right order: you first refer to Fig 5.

p8 l 260: you assume in Fig 5 that settling velocity is varying linearly and not vice versa. How valid is this assumption?

p8 l262: 'sediment particles': do you mean 'mineral particles'?

p9 l277: 'former studies': which one?

p9 278-280: "The correlation is, however, rather poor . . . and can be explained by . . .."

and af course of turbulence. SPMC and turbulence determine the settling velocity.

p10 l308-309: "This suggest that the region can be considered as a transition zone, hindering mineral particles to escape further off-shore." What about the effect of deeper water depth that result in a dilution of the SPM? Or the fact that the transition zone is further off-shore and thus tidal currents and tidal-current ellipses changes? Is what you have observed (gradient in SPMC) not also related to these processes? Maybe that this is not relevant for the study, but I am intrigued by features like the East Anglia plume that extend far into the North Sea (up to the German Bight), and that are not restricted to certain turbulence regimes or 'transition zone'. Do you see the East Anglia plume in your data: higher SPMC further off shore?

p11 l 363-364: "Hence, the ability for nutrient retention is diminished and would lead to generally lower nutrient concentrations in similarly affected Wadden Sea regions." You have supposed similar parameters in whole the Wadden Sea. Is this correct in view of the different behavior of the Sylt-Romo basin?

p11 l367: I don't understand what the physical basis is to link gradients in SPMC to the spatial distribution of ws.

Figure 2: What is GETM? Figure 4: is not really convincing as it is a log-log plot with only few very low SPMC values. Figure 7: Meris image: use a more appropriate scale (e.g. starting at 1 mg/l in staed of 0.1 mg/l).

---

## Referee Comment (RC2) · Anonymous Referee #3 · 23 May 2016

General Comments

The authors analyse an extensive data-set of vertical SPM concentration profiles in the German Bight. They analyse this data-set to reveal the overall cross-shore spatial structure in settling velocities of (aggregates of) particulate matter. They find a maximum in estimated settling velocities in a 15-20m depth zone. The authors put forward two alternative possible explanations for this maximum. First, this could be the zone with optimal balance between turbulence controlled floc-formation and break-up, resulting in the largest flocs and hence the largest settling velocities. Second, they hypothesize that the composition of SPM could vary along a cross-shore gradient. They demonstrate that a functional relation linking settling velocity to primary particle size,

floc size, fractal dimension of flocs and relative amount of organic matter in SPM can also show a maximum when all explanatory variables are allowed to vary linearly along a cross-shore gradient. They discuss these observations in terms of the biogeochemical cycling of the German Wadden Sea and hypothesize that along shore differences in the spatial variability of settling velocities could explain regional differences in nutrient cycling and eutrophication status. Although speculative, the discussion of potential implications is interesting. To interpret the observations the authors make use of existing hydrodynamic model simulations, i.c. to estimate the vertical eddy diffusivity and energy dissipation rate. As such they are able to present a 2D data set of SPM concentrations and settling velocities as a 1D relation with energy dissipation rate. This is a nice example of the combined use of in-situ observations and model simulations to retrieve new and insightful information from a system. My major comment is regards the discussion section, which I find short and incomplete. On a first reading it is not clear that the authors are discussing two alternative explanations for the observed maximal settling velocity. This could be made more clear. Neither of both possible explanations of a maximal settling velocity is properly discussed on how realistic it is. No attempt is made to compare the epsilon values at which the a maximum occurs to other research on floc formation and break-up. Is the reported epsilon range of maximal settling velocities indeed those that lead to maximal floc size, and under which circumstances? Similar for the results of the conceptual model. Is there any additional support that the different explanatory variables increase linearly over the same distance across the shore? Only the end-members are shortly introduced in relation to other work. It is unclear whether the location of the maximum in the conceptual model can indeed be in the same zone as the observed maximum settling velocity, and under which assumptions. Consequently, it should be discussed which of both potential explanations the authors find the most important, or whether both are deemed equally important. Finally, the authors make no attemt to discuss other explanation for the spatial variabiliy in eutrophication status of the Wadden Sea and why the proposed mechanism here is the most plausible, as seems to be implied by their manuscript. Partially because of

the incomplete discussion, I find the conclusions often speculative, e.g.

P11 L21-22 "Algae with their seasonality seem to be strongly involved". There is no evidence in the underlying paper supporting this statement. On the contrary, the authors demonstrate that also turbulence controlled flocculation and break-up can explain the existence of a maximal settling velocity. In that case, algae are not involved in the establishment of the coastal transistion zone. Overall, I think the authors will be able to address most of the comments with a few additional paragraphs and rewriting the existing text. Therefore I recommend Minor Revisions for this paper.

Technical points

The draft needs many textual and technical improvements, and different parts of the methods need further clarification among which:

p4 L25 eq 1 Clarify: z-axis pointing downward

p4 L28-30. The text motivates why k is taken constant over time (i.e. because settling time scales are much longer than the tidal time scale.). This does not motivate why k can be taken constant over depth, which is the essential assumption to arrive at solution (2). This should be motivated separately. In fact, equation (1) only makes sense as a tidally averaged balance in which $k_v$ is already taken constant in time. Otherwise dC/dt would not vanish, and horizontal advection would not be zero.

P4 L30 I think the authors are wrong here: the exponential solution of equation (1) assumes zero net fluxes across the vertical boundaries. Otherwise source terms would show up in the solution.

P4 L30 rephrase: "Assuming zero fluxes across the vertical boundaries and a vertical diffusivity that is constant over depth, equation (1) can be integrated over a water column with arbitrary height $H_p$. This results in an exponential concentration profile. The unknown integration constant can be expressed as a function of the (unknown) average concentration of the profile as: "

P4 L30: motivate why you express the exponential profile as a function of the unknown average concentration over the profile. You could as well express it as a function of the concentration at a specified depth (e.g. at the top or bottom of the concentration)

P4 L31 $C_m$ is not defined

P4 L31 Be precise in your notation: in this formula <.> denotes a depth average over concentration profile, while later in the text, you state that <.> means an ensemble average.

P5 L2-4 It would be helpful to specify the turbulence model that was used in the hydrodynamic simulations and how vertical turbulent diffusivity was estimated from it.

P5 L2 required → requires

P5 L 8 remove comma between "both" and and "observed"

P5 L13-15 How do you motivate the choice for the limits of density gradient standard deviation. You state that they are "somewhat arbitrary", which implicates that they are also "somewhat motivated". Please do so.

P5 L14 See above: here <.> denotes averaging over ensembles. Be precise.

P5 L14 Clarify: What do you mean by ensemble average, is it the average over an epsilon-bin?

P5 L17-27. I have difficulty to believe this is a correct way to split the vertical SPM profile in subsections. To my best knowledge, density stratification in the German bight is related to water temperature. SPM concentrations and their vertical gradients in the German bight seem too low to me to induce a significant impact on densities and induced turbulence dampening. Large density gradients are in itself the cause of reduced turbulence, whatever the cause of the density gradient. This indeed causes a strongly reduced turbulent diffusivity in the deeper waters compared to the surface waters. As I understand it, this is the principal reason why profiles need to be split in 2

sections on which 2 exponential profiles with constant $k\_v$ are fitted, one representative for "deep" waters and the other for "surface" waters . Therefore, the authors should motivate why they not just split the profiles at the level where maximal density gradient exists, and what the difference would be with their method.

P5 L23. Here again: what kind of averages are denoted with <.>. It seems each single profile was considered, thus <.> would mean averaging over depth.

P5 L34 It took me a while to find what F means. Perhaps this can be clarified here again

P7 L8. Rephrase: "The sensitivity of $w\_s$ to changes in each parameter was assessed by varying each parameter separately while keeping the other parameters at their typical values for coastal waters, I.e...."

P7 L11 "Vertically and temporally averaged model-derived ..." Over which period was the temporal averaging

P7 L26 & Fig5 I find the tilted figure a bit difficult to read, why not put it straight up?

P11 L4 Change "rises" to "raises"

---

## Author Comment (AC1) · 20 Jun 2016

**Reply to anonymous referee #2**

Dear referee,

thank you very much for your really valuable and constructive comments on our manuscript! In the following, a detailed listing of answers to your general and specific comments is given. Your comment is set in italic and respective answers are set normal.

General comment:

*“The manuscript aims to present new evidence to understand SPM dynamics in coastal seas, especially the mechanism that sustains the net-transport of SPM towards coastal systems, such as the Wadden Sea. The authors have used an approach that indirectly calculates the settling velocity in a cross-shore direction by using measured SPMC profiles and modelled turbulence data. The approach is original and the results are convincing. However, I am not in favor to call the obtained results an ’evidence’, as the methods is based on a lot of assumptions that are not providing ’evidence based data’. Some of the assumptions used are further speculative and thus not supporting evidence. The latter would, e.g. be the case if settling velocity would have been measured directly. Nevertheless, even without direct evidence, the manuscript remains valuable and will surely inspire other scientists to look for similar mechanism in other coastal regions or to adapt existing monitoring programs to collect more evidence based data to prove the hypothesis. My major comment is therefore to not argue that you have found evidence to accept the hypothesis. Change the title accordingly and weaken the conclusions. As I have also some specific comments (see below) that are going further than minor changes I recommend to accept the manuscript only after major revisions.”*

Changed the title to “Maximum sinking velocities of suspended particulate matter in a coastal transition zone”

We weakened the conclusions by replacing ’evidence’ with ’strong indication’ and also weakened the retention capability by including ’processes would have the potential to retain fine sediments and nutrients in coastal areas’. To make the importance of turbulence features more clear in the evolution of the transition zone, we additionally included a semi-clause on the relevance of the energy dissipation rate intensity.

Specific comments:

*"SPM concentration or SPMC: try to be consistent. Sinking velocity or settling velocity: idem"*

We are now using SPMC and sinking velocity throughout the whole text.

*"p2 l27-28" Cohesive sediments and POM can undergo aggregation and fragmentation processes that change transport properties and thus their sinking velocity". Change the sentence into: flocculation changes the settling velocity and thus the transport properties and not the opposite way.*

Order changed.

*In previous sentence you make the difference between cohesive and non-cohesive minerals, thus between clays and the other minerals such as quartz or carbonates. Do you have evidence that only the cohesive minerals are involved in flocculation? What about very fine quartz or carbonates that can due to electric charges or the presence of specific organic molecules (EPS) be involved in flocculation?"*

We changed the sentence to: "Fine-grained minerals of sizes typically up to 8$\mu$m (Chang et al. 2006) and POM can undergo aggregation and fragmentation processes that change sinking velocity and thus transport properties."

*"p2 l37-39: "In shallow waters near the coast, where turbulence and thus resuspension are high, SPM concentration is usually enhanced and dominated by mineral particles with high densities". Better: The SPM concentration consist of flocs that are composed mainly of mineral particles. p2 l39-40: "By contrast, in deeper off-shore regions, SPM concentration is comparably low and consists to a higher extent of POM with low*

*densities." Better: . . . SPM concentration is lower and the flocs are looser and more organic."*

Changed accordingly.

*"p2 l51: what do you mean by 'restructuring processes'?"*

We now give a reference to Becker et al 2009, who investigated restructuring (e.g. resulting in compaction – less fractal flocs) process under shear stress theoretically.

*"p3 l67: add a reference for the 'typical power law relation'"*

We included the reference: 'Dyer 1989'.

*"p3 l94: an average depth of 80 m is not what I would call 'shallow'. Further, the water depth in the German Bight seems to be much less than 80 m from Fig 1."*

We changed the sentences to: "The surveyed German Bight (Fig. 1) is located in the south-east of the North Sea and features a typical depth of about 30 m to maximal 50 m. The North Sea is a shallow shelf sea with an average depth of 80 m . . . ". Compared to typical definitions of continental shelf with about 200 m water depth and for the Antarctic continental shelf of even about 400 m, an average depth of the North Sea with 80m is shallow.

*"p4 l99: The Rhine is not the only source of SPM and nutrient in the southern North Sea: see the general circulation pattern that brings Channel water into the North Sea and all the subsequent sources. What about the East Anglia plume that extends towards the German Bight?"*

We agree that the East Anglia plume occasionally extend towards the German Bight. We now state this in the description of the study area. However, during the cruises no

SPMC increase was found in the outer region of the surveyed study area which could be attributed to the East Anglia plume. We checked the scanfish data again in this respect. For surface waters, this is also visible in Fig. 7, upper panel. Additionally, most of our observations stem from regions that are usually not affected by the East Anglia plume.

*"p4 l110-113: I don't understand the hypothesis of 'line-of no-return"'*

We added the sentence: "The line-of-no-return is conceptually described as the imaginary line beyond which particles escape coastal trapping mechanisms such as e. g. density gradient-driven undercurrents (Postma 1984)." to give more explanation.

*"p4 l119: The cruises are all in spring/summer period. Would the hypothesis of maximum settling velocity also be valid in winter? Do you have winter data?"*

We added the sentence: "No winter measurements were carried out." Hence, the validity of the hypothesized/found zone of maximum sinking velocity has still to be explored for winter. We think that we made it clear in the conclusions that more investigation on seasonal patterns are required.

*"p4 l123: sampling rate of 11 s-1: better 11 Hz"*

Changed.

*"p4 l124: conductivity or specific conductance. I am not sure that the term 'specific conductivity' is used."*

Changed to: "specific conductance"

*"p4 l126: Is a Seapoint turbidity meter appropriate to be used in the high turbidity coastal areas where you say that SPM concentration is > a few 100mg/l."*

Typically, SPMC of much lower values were measured. As shortly described, obvious faults – such as clipping due to too high turbidity values would have been removed during post-processing of the data. In addition, the Seapoint turbidity meter we are using has a range up to 500 FTU, a value that was never reached during our surveys.

*"p5 l147: "If we assume that the sinking time scale is larger than the tidal period". How valid is this assumption? The time scale of a tidal cycle is about 12.5 h (or if you consider ebb/flood: about 6h), and of the sinking time scale about 3-6h (w=1 mm/s in 10m water depth: 3h, ws=0.5 mm/s in 10 m water depth: 6h). This seems to me quiet similar. What are the consequences of this assumption on the results?"*

We assume that the effects are of second order. Would we relax the assumption on the sinking time scale, we had to assume a parabolic viscosity profile. This would result in an SPM profile with an power law shape $a * \mathrm{SPMC}^b$ instead of the exponential profile. However, the introduction of a parabolic viscosity profile would introduce a new unknown constant, the reference concentration in the bottom boundary layer. Despite this theoretical considerations, we still believe that the differences are small. Since we do a local fit of the SPM profile, the the difference between a power law fit and an exponential fit are smaller than the errors introduced by the parameter fitting. Moreover, the introduction of the reference concentration for the power law fit would introduce a further source of uncertainty and an additional free fitting parameter.This free parameter would be difficult to bound, since the measurements did not resolved the bottom boundary layer.

Concluding, although we know that the assumption of an exponential concentration profile provides an upper bound on the sinking velocity, it is a good trade of between accuracy and over-fitting of free parameters.

*"p5 l150: $C_m$ is not defined. Is this the depth dependent SPMC?"*

Yes, it is the analytical model description of the theoretically expected vertical SPMC

distribution. We added $C_m$ after 'analytical model'.

*"p5 l160ff: How big is the difference between model results and observations. It would help a lot to better understand the procedure if you would show examples of the fitting, interpolations etc."*

We now provide an exemplary figure, where the different steps of the data processing with respect to model-observations comparison, profile splitting and analytical model fitting are shown. Additionally, we added a sentence in the text to point to the figure.

*"p6 l174 On what is this criterion (< 0.0005 kg m-3 m-1) based see line 168 where it is 0.015."*

We noted in the description of the Monte-Carlo-type simulation at the end of Sec. 2.3 that threshold values were selected upon visual inspection criteria. To have direct correspondence, we now added: "Since the critical value was set by visual inspection, it was also considered in the Monte-Carlo-type simulation described below."

*"p6 l175-176: "The co-occurrence of strong gradients in $\sigma_T$ and SPMC can indicate dampening of turbulent mixing and potential particle properties' changes." Is it always the case that strong gradients in sigma T are co-occurring with those in SPMC? What if only one of the two parameters has strong gradients?"*

A vertical gradient of SPMC is what we expect to see, while a strong gradient in $\sigma_T$ can, but must not necessarily, imply potential particle property changes. We added the sentence "While we generally expected vertical gradients in SPMC, strong vertical gradients in SPMC potentially reflect weak mixing and are thus probably an additional indicator for changes in turbulence intensities." and included an expanded explanation for our splitting approach. See also the text and the comment to reviewer #3.

*"p6 line 189: On what is the cost function value of 0.05 based?"*

We added ", chosen by visual inspection,".

*"p6 line 190: the variable F is not defined."*

It is/was defined in Sec. 2.2 l.127 of the reviewed version of the manuscript. It is fluorescence. We now nevertheless introduce it here again and clearly state: "…SPMC, and fluorescence to SPMC ratio (F/SPMC) …"

*"Why is it necessary to bin the data. If I understand well the methods than the data have already been interpolated on 5 cm resolution."*

For every profile, for which one $w_s$ value was found, variables SPMC, etc. were vertically averaged. Afterward, those value and the according $w_s$ were binned with respect to epsilon, which relates to turbulent shear as an important driver in flocculation processes. We rephrased the sentence to: "The variables SPMC, and fluorescence to SPMC ratio (F/SPMC) were vertically averaged for the respective (sub-) profiles. Afterward, variables were binned with respect to modeled $\epsilon$ ..." We hope, the procedure becomes more clear now.

*"p7 l 209: "be described operationally as fractal dimension-like": what do you mean by 'operationally', skip '-like"'*

Both, "operationally" and "-like" were written to state that fractal dimension is here not meant in a strict mathematical way – so that there exists a cutoff, especially for very small particles reaching the limit of the primary particle diameter size. Since this is a detail that does not concern our results and the conceptual model, we simplified the sentence and skipped both, "operationally" and "-like".

*"p8 l246-249: I don't agree on the potential significance depicted by F:SPMC ratio. Fig 3b shows in fact that F is increasing towards the high turbulent areas (towards the*

*coast thus). As SPMC is however stronger increasing than F the ratio is dropping. The highest alga concentrations are generally found in the nutrient rich, high turbid coastal areas."*

We agree that the total amount of biomass tend to increase towards the coast. However, the relative weight/weight ratio of POM/SPM typically decreases towards the coast (so high loss on ignition in the deeper regions compared to low loss on ignition in the shallow, turbid regions – this is ubiquitously observed and also conceptually introduced in our introduction). Since on first approximation, fluorescence F is proportional to POM, it should be eligible to use F/SPM ratio as proxy for loss on ignition. To clarify our assumption, we rephrased and added a sentence: "Under the assumption that fluorescence can be applied as a proxy for POM, the potential significance of algae and their products for particle composition is depicted by the mean ratio between measured fluorescence and SPMC. High F/SPMC ratios likely indicate rather organic-rich particles, while low F/SPMC ratios potentially indicate rather mineral particle-dominated flocs. "

*"Fig 4 is not in the right order: you first refer to Fig 5."*

Changed.

*"p8 l 260: you assume in Fig 5 that settling velocity is varying linearly and not vice versa. How valid is this assumption?"*

We do not assume that sinking velocity varies linearly, we only assume linear changes of parameters that enter the calculation of the sinking velocity according to Eq. 8. Since for parameters omega, $\rho_p$, and $\rho_o$, sinking velocity has a linear dependency (and if $d_f = 2$, for $D_p$ and D as well), it is obvious that $w_s$ also varies linearly. When considering all linearly changing parameters for the calculation of sinking velocity $w_s$, then $w_s$ shows a maximum. Anyways, there was the word 'results' too much, we removed it.

*“p8 l262: ‘sediment particles’: do you mean ‘mineral particles’?”*

Changed from ’sediment’ to ’mineral’.

*“p9 l277: ‘former studies’: which one?”*

We give now a reference to Dyer 1989.

*“p9 278-280: “The correlation is, however, rather poor . . . and can be explained by . . . ” and of course of turbulence. SPMC and turbulence determine the settling velocity. ”*

We now explicitly mention turbulence, while we do not include the added sentence since it appears similarly a few lines later at the end of the Sec. 4.1.

*“p10 l308-309: “This suggest that the region can be considered as a transition zone, hindering mineral particles to escape further off-shore.” What about the effect of deeper water depth that result in a dilution of the SPM? Or the fact that the transition zone is further off-shore and thus tidal currents and tidal-current ellipses changes? Is what you have observed (gradient in SPMC) not also related to these processes? Maybe that this is not relevant for the study, but I am intrigued by features like the East Anglia plume that extend far into the North Sea (up to the German Bight), and that are not restricted to certain turbulence regimes or ‘transition zone’. Do you see the East Anglia plume in your data: higher SPMC further off shore?”*

We thank the reviewer for this aspect and now acknowledge the dilution effect and potential effect of parallel to the shore line occurring currents by introducing the follwoing sentence in the discussion: “Generally, dilution of SPMC occurs due to cross-shore wise increasing water depth and locally occurring currents parallel to the coast potentially confine horizontal SPMC distribution to the near-coast region (Staneva et al. 2009).”

We re-checked all campaigns and never observed a pronounced increase in the outer German Bight in the region of the so-called 'Entenschnabel'. The East Anglian Plume is a transient feature at the outer parts of the German Bight and was, in terms of SPMC, not observed in our study. The latter can, for surface waters, also be inferred from the MERIS image (Fig.7).

We added the sentence: "To simplify, consider the course from the coast to open waters. Once formed, relatively dense, fast sinking flocs, whose properties are adapted to the transition zones' turbulence level, would easily settle out of the water column once transported offshore. That is because turbulence becomes too weak to break those aggregates apart and to retain them in suspension. Thus, only loose, organic-rich particles are kept suspended in the water column while mineral-rich particles tend to settle to the near-bottom waters."

*"p11 l 363-364: "Hence, the ability for nutrient retention is diminished and would lead to generally lower nutrient concentrations in similarly affected Wadden Sea regions." You have supposed similar parameters in whole the Wadden Sea. Is this correct in view of the different behavior of the Sylt-Romo basin?"*

Maybe we misunderstood the comment, but that is exactly what we discuss: the Sylt-Rømø-Bight likely features a lower turbulence levels than other tidal inlets, also visible in Fig. 2, which is reflected in the mapped $w_s$ and thus the retention capability for nutrients is diminished. Thus the concept of the transition zone can help to understand the lower nutrient concentrations in the Sylt-Rømø Bight compared to the East Frisian islands since the transition zone is not as pronounced along Sylt-Rømø as along the East Frisian islands. We rephrased a sentence to: "Hence, the ability for nutrient retention is diminished and would contribute to the lower nutrient concentrations compared to other Wadden Sea regions", which hopefully clarifies our view.

*"p11 l367: I don't understand what the physical basis is to link gradients in SPMC to*

*the spatial distribution of ws."*

We added the sentence: "Flocs with high $w_s$, that are likely mineral-dominated, rapidly sink out of the water column and thus lead to strong cross-coastal diminishing of SPMC." and believe that the aforementioned description on comment *"p10 l308-309"* additionally help to understand the relevance of $w_s$ for cross-shore SPMC gradients.

*"Figure 2: What is GETM?"*

Caption changed to: "Map of hydrodynamic model-calculated time- and depth-averaged energy dissipation rate epsilon for the times of the cruises." (GETM is the general estuarine transport model that was used in the study of Gräwe et al. 2015)

*"Figure 4: is not really convincing as it is a log-log plot with only few very low SPMC values. Figure 7: Meris image: use a more appropriate scale (e.g. starting at 1 mg/l instead of 0.1 mg/l)."*

We now chose a different binning so that more bins are in the low SPMC range. The scale for Fig. 7 has changed accordingly

Please also note the supplement to this comment:
http://www.biogeosciences-discuss.net/bg-2015-667/bg-2015-667-AC1-supplement.pdf

**Supplement:**

[revised manuscript text omitted]

10    cal solution to observed SPMC profiles  requires information on correspondent $k_{\mathrm{v}}$. These, and additionally energy dissipation rate $\epsilon$ and water density $\sigma_T$ were obtained from hydrodynamical simulations of Gräwe et al. (2015), based on a 1 nautical mile numerical model of the North Sea and the Baltic Sea. In the vertical, 42 terrain-following levels were used. Vertical mixing is parametrized by means of of a two-equation $k - \epsilon$ turbulence model coupled to an algebraic second-moment closure (Canuto et al., 2001). The implementation of the turbulence module is done via the General Ocean Turbulence Model

15    (GOTM; Umlauf and Burchard, 2005). The hydrodynamic parameters were stored as 2 hourly snapshots.

Hydrodynamic model results for $k_v$, $\epsilon$, and $\sigma_T$ were linearly interpolated in space and time to extract a corresponding value for every measured data point. Even though state-of-the-art hydrodynamical models perform generally very well, they may locally exhibit discrepancies to observations. To eventually discriminate for lacking congruency, both  observed and modeled $\sigma_T$, were interpolated on a common vertical grid of $\Delta z = 0.05\,\mathrm{m}$ and filtered by applying a least-square straight

20    line fit to the data and a 'natural' cubic spline interpolant. The latter had a weight of $80\,\%$ to produce a smooth interpolation curve $\sigma_T'(z')$. Subtracting the respective vertical mean resulted in profiles $\widetilde{\sigma_T}$ for both observed and modeled data. We constrained our further analysis to casts satisfying two criteria. First, after subtraction of modeled $\widetilde{\sigma_T}$ from observed data, the standard deviation should not exceed $\mathrm{std}(\Delta\widetilde{\sigma_T}) \leq 0.015\,\mathrm{kg\,m^{-3}}$. Second, the density gradients defined as $\delta_z\sigma_T = (\langle\sigma_T'(\mathrm{last\,meter})\rangle - \langle\sigma_T'(\mathrm{first\,meter})\rangle)/H$, where $\langle x\rangle$ generally represents the  mean of a variable $x$ (here specifically

25    the vertical mean), should not exceed a discrepancy of $\Delta\delta_z\sigma_T \leq 0.015\,\mathrm{kg\,m^{-3}\,m^{-1}}$ between observed and modeled data. Both limits for $\mathrm{std}(\Delta\widetilde{\sigma_T})$ and $\Delta\delta_z\sigma_T$ were applied to select for similar vertically structured observed and modeled density profiles. The chosen values, however, were somewhat arbitrary and therefore considered in a Monte-Carlo-type simulation (see below).

ScanFish cruises covered well-mixed coastal, but partly also stratified waters in the inner German Bight. A consistent approach to retrieve $w_s$ was required to meet both conditions. We chose to split casts into subprofiles, if needed, and to fit

30    the single analytical solution, Eq. (2), to (sub-) profiles. If observed $\delta_z\sigma_T < 5\cdot10^{-4}\,\mathrm{kg\,m^{-3}\,m^{-1}}$, the whole profile was used for fitting, otherwise the cast was split. Since the critical value was set by visual inspection, it was also considered in the Monte-Carlo-type simulation described below. Strong gradients in $\sigma_T$ are typically considered to indicate dampening of

turbulent mixing. The squared buoyancy frequency ($N^2$)

$$N^2 = \frac{g}{\rho} \frac{\partial \rho}{\partial z} \quad , \tag{3}$$

where $g$ denotes the gravitational acceleration constant, is related to the vertical eddy diffusivity $k_v$ by (Osborn and Cox, 1972)

$$k_v = c \frac{\epsilon}{N^2} \tag{4}$$

5     with the current standard value for $c = 0.2$ (Lindborg and Brethouwer, 2008) . A strong vertical sea water density gradient thus reflects low turbulent diffusion, which would imply to split profiles in stratified waters at the maximum density gradient(s). However, as recently discussed by Franks (2014) , mixed layer depths defined by density gradients are a rather inadequate proxy for turbulence intensity. Since particle properties such as size and density are a function of shear rate and SPM components, it is reasonable to expect them to differ with turbulence intensity in a vertical profile leading to different sinking velocities. For

10     example, Leipe et al. (2000) reported vertically varying mean aggregate sizes for the Baltic Sea. While we generally expected vertical gradients in SPMC, strong vertical gradients in SPMC potentially reflect weak mixing and are thus probably an additional indicator for changes in turbulence intensities. The co-occurrence of strong gradients in $\sigma_T$ and SPMC can  thus indicate changes in turbulence intensity and potential particle properties' changes. Splitting at these points allow us to apply the analytical SPMC model (Eq. (2)), where we assumed a rather homogeneous turbulent

15     diffusion by using the vertically averaged $k_v$. Thus, end points of subprofiles were set, where regions in the profile with

$$\left| \frac{1}{C'} \frac{\partial C'}{\partial z'} \cdot \frac{1}{\sigma_T'} \frac{\partial \sigma_T'}{\partial z} \right| > \left\langle \left| \frac{1}{C'} \frac{\partial C'}{\partial z'} \right| \right\rangle \left\langle \left| \frac{1}{\sigma_T'} \frac{\partial \sigma_T'}{\partial z} \right| \right\rangle \tag{5}$$

occur, and reach their maximum, where $\langle x \rangle$ again denotes the vertical average. $C'$ is the SPMC smoothing spline-interpolated analogously to $\sigma_T$. Start points of the subprofiles were either the first data point from surface or where regions defined by Eq. (5) end with increasing depth. Only subprofiles longer than $4\,\mathrm{m}$ were considered in the further analysis.

[revised manuscript text omitted]

5 (according to Maggi, 2009) and $\rho_o = 1100\,\text{kg}\,\text{m}^{-3}$ (as an average value of the range given in Fettweis, 2008). The water density and the dynamic viscosity are set to $\rho = 1000\,\text{kg}\,\text{m}^{-3}$ and $\mu = 0.001\,\text{kg}\,\text{m}^{-1}\,\text{s}^{-1}$, respectively.  The sensitivity of $w_s$ to changes in each parameter was assessed by varying each parameter separately while keeping the other parameter parameters at their typical values for coastal waters, i.e. for an organic-rich aggregate with an assumed diameter of $D = 100\,\mu\text{m}$, and $\omega = 0.5$,

10 $D_p = 4\,\mu\text{m}$, and $d_f = 2$.

**3 Results**

**3.1 Cross-coastal gradients**

Energy dissipation rates $\epsilon$ generally possess spatio-temporal variability. Most relevant for this study, vertically and temporally averaged model-derived $\epsilon$ exhibit a cross-shore gradient from the inner German Bight towards the coast. The temporal

15 averaging was carried out for the times of the cruises while accounting for the length of the tidal cycle. Values range between $\epsilon \approx 10^{-9}\,\text{m}^2\,\text{s}^{-3}$ and $\epsilon \approx 10^{-4}\,\text{m}^2\,\text{s}^{-3}$, respectively (Fig. 3).

Along this range, mean SPMC first increased from values below $1\,\text{g}\,\text{m}^{-3}$ to approximately $10\,\text{g}\,\text{m}^{-3}$. Above $\log_{10}(\epsilon) = -5.5$, however, mean SPMC moderately decreased to approximately $6\,\text{g}\,\text{m}^{-3}$ and increased again to $10\,\text{g}\,\text{m}^{-3}$ with further increasing $\epsilon$ (Fig. 4 A) towards the coast.

20  Under the assumption that fluorescence can be applied as a proxy for POM, the potential significance of algae and their products for particle composition is depicted by the mean ratio between measured fluorescence and SPMC.  High F/SPMC ratios likely indicate rather organic-rich particles, while low F/SPMC ratios potentially indicate rather mineral particle-dominated flocs. The F/SPMC ratio shows rather high variability for $\log_{10}(\epsilon) < -6$ (Fig. 4 B). With further increasing $\epsilon$, the ratio drops and levels at approximately a fourth of the open German Bight value for $\log_{10}(\epsilon) > -5$.

25 Average sinking velocities $\langle w_s \rangle$ show very low values of order $10^{-6}\,\text{m}\,\text{s}^{-1}$ to $10^{-5}\,\text{m}\,\text{s}^{-1}$ in regions of $\log_{10}(\epsilon) < -7.5$ (Fig. 4 C). At higher $\epsilon$, $\langle w_s \rangle$ increased significantly reaching a maximum of about $7 \cdot 10^{-4}\,\text{m}\,\text{s}^{-1}$ around $\log_{10}(\epsilon) \approx -5.5$. This maximum $\langle w_s \rangle$ around $\log_{10}(\epsilon) \approx -5.5$ coincided with the increase of SPMC, and the drop in fluorescence to SPMC ratio. With further increasing $\epsilon$, $\langle w_s \rangle$ decreased again to values of $\langle w_s \rangle \approx 3 \cdot 10^{-4}\,\text{m}\,\text{s}^{-1}$. The Monte-Carlo-type simulation to estimate parameter uncertainties exhibits the same pattern, expressed as $\sigma$ and $2\sigma$ confidence levels, $68\,\%$ and $95\,\%$ respectively

30 (Fig. 4C), around their ensemble mean, and thus underpins the results of $\langle w_s \rangle$ along $\epsilon$.

**3.2 Conceptual model**

 Sinking velocity of an average particle (as parametrized in Sec. 2.4) changes, after variation in single parameters along the transect from open to coastal waters, in an almost linear way (Fig. 5, upper four panels). Flocs would sink less rapid because of decreasing diameter of the aggregate or the primary particles in the coastal region. By contrast, increasing amount of  mineral particles and fractal dimension would both lead to increased $w_s$. As an overall effect, when considering all parameters, $w_s$ reaches a maximum in between the near-shore and open waters (lowest panel of Fig. 5).

**4 Discussion**

**4.1 Sinking velocity on a gradient of prevailing energy dissipation rate**

Sinking velocities have been determined along a cross-shore transect in the German Bight defined by the prevailing $\epsilon$. For the reconstruction of $\langle w_s \rangle$, we had to assume congruency between *in situ* measurements and hydrodynamic model results within a range defined by the applied filters. The estimated $\langle w_s \rangle$ of order $10^{-6}\,\mathrm{m\,s^{-1}}$ to $5 \cdot 10^{-4}\,\mathrm{m\,s^{-1}}$ in low and high energy dissipation regions, respectively (Fig. 4), compare very well with former *in situ* studies in the German Bight and adjacent areas. Puls et al. (1995) found $w_s \approx 10^{-6}\,\mathrm{m\,s^{-1}}$ to $2 \cdot 10^{-5}\,\mathrm{m\,s^{-1}}$ for the open German Bight, a low $\epsilon$ region, and Pejrup and Mikkelsen (2010) reported median $w_s$ of $1 \cdot 10^{-4}\,\mathrm{m\,s^{-1}}$ to $11 \cdot 10^{-4}\,\mathrm{m\,s^{-1}}$ for the Rømø and Høyer Dyb, Danish Wadden Sea where $\epsilon$ is comparably high. Our estimated $\langle w_s \rangle$ are also well in the range found in other regions, e. g. in Chesapeake Bay (Gibbs, 1985) or at the Belgian Coast (Fettweis and Baeye, 2015). In agreement with former studies (e. g. compiled in Dyer, 1989), obtained $\langle w_s \rangle$ increase with increasing $\langle \mathrm{SPMC} \rangle$ (Fig. 6), which gave further confidence in the methodological approach. The correlation is, however, rather poor compared to former local studies and can be explained by the heterogeneity of the German Bight system, seasonal effects and the intrinsically high variability of sinking velocities (Fettweis, 2008) and of course of turbulence (Pejrup and Mikkelsen, 2010). With respect to prevailing $\epsilon$, the significant maximum of $\langle w_s \rangle$ at $\langle \epsilon \rangle \approx 10^{-5.5}\,\mathrm{m^2\,s^{-3}}$ can be explained by the balancing between aggregation and fragmentation. Both processes are controlled by turbulent shear, generated by energy dissipation, SPM volume concentration, and adhesion properties of particles involved. Previous flocculation modeling studies pointed to a formation of a $w_s$ maximum along $\epsilon$ (Winterwerp, 1998; Pejrup and Mikkelsen, 2010). Our findings of a maximum in $\langle w_s \rangle$ between $\epsilon \approx 10^{-6}\,\mathrm{m^2\,s^{-3}}$ and $10^{-5}\,\mathrm{m^2\,s^{-3}}$  which translates to shear rates of about $1\,\mathrm{s^{-1}}$ to about $3.2\,\mathrm{s^{-1}}$ for $\nu = 10^{-6}\,\mathrm{m^2\,s^{-1}}$, should thus be comparable to former theoretical studies. For example, in a field work-based modeling approach Pejrup and Mikkelsen (2010) found the maximum of sinking velocity to occur under higher shear rates of about $8.5\,\mathrm{s^{-1}}$. By contrast, Maerz et al. (2011) calculated highest mean sinking velocities of about $5 \cdot 10^{-4}\,\mathrm{m\,s^{-1}}$, similar to our findings, for a jar test device-based lab experiment with natural SPM during shear rates of about $1\,\mathrm{s^{-1}}$. Given the underlying uncertainties in calculating the shear rate in both cases, our study thus agrees well with former theoretical studies. Hence, as already suggested by Dyer (1989) and Pejrup and Mikkelsen (2010), in addition to SPMC, turbulent shear can be regarded as the major determinant for $w_s$.

**4.2 Sinking velocity as a result of SPMC, composition and a turbulence gradient**

Our study additionally suggests a change in particle composition with $\epsilon$ as proxied by the fluorescence to SPMC ratio. With $\epsilon$, SPMC increases towards the coast and mineral fraction becomes dominant compared to organic particles at low $\epsilon$ (Eisma and Kalf, 1987). Hence, density and likely other physico-chemical properties of primary particles such as size and
5    adhesion change accordingly and thus influence $w_s$ along a cross-shore transect. Our conceptual model illustrates the effect of varying particle properties on $w_s$ independent of the observations used in our data analysis. However, the course of the parameters implicitly presumes certain environmental conditions, as e. g. size of particles is, among others, a function of SPMC and shear. Single parameter changes in the conceptual model resulted in mostly linear responses in $w_s$, while considering all parameters led to a maximum in $w_s$ on the conceptual cross-shore transect, in alignment with our findings in the data analysis.
10   The assumption of linearly changing parameters is a first order approximation. Differently and non-linear changing parameters would lead to a transformation of the maximum sinking velocity zone such as a shifting and / or stretching (not shown). Former model studies have described for a given shear regime how varying particle properties such as particle adhesion lead to changing particle sizes and thus sinking velocity (e. g. Maerz et al., 2011) . Hence, higher particle adhesion would translate to larger particles and stronger fragmentation resistance which would allow particles sink out of the water column closer to
15   the coast under higher turbulence intensities. This implies that spatial changes in SPMC and composition, and accordingly physico-chemical properties of particles, potentially affect the location of the transition zone. As discussed below this probably happens on a seasonal time scale due to the modulation of the transition zone by phytoplankton exudations which requires further investigations. While the cross-shore distribution of $w_s$ is predominantly controlled by prevailing shear, the conceptual model indicates  a potential additional relevance of physico-chemical properties for the formation of the high sinking
20   velocity zone.

**4.3 Implications of a cross-coastal maximum of sinking velocity for biogeochemical cycling in the coastal zone**

The region of $\log_{10}(\epsilon) \approx -5.5$ with highest $\langle w_s \rangle$ coincides with a zone located off the coast at depth of about $15\,\mathrm{m}$ to $20\,\mathrm{m}$. It is accompanied by a strong gradient in  SPMC and likely composition as depicted by the F/SPMC ratio. This  suggests that the region can be considered as a transition zone, hindering mineral particles to escape further off-
25   shore. To simplify, consider the course from the coast to open waters. Once formed, relatively dense, fast sinking flocs, whose properties are adapted to the transition zones' turbulence level, would easily settle out of the water column once transported offshore. That is because turbulence becomes too weak to break those aggregates apart and to retain them in suspension. Thus, only loose, organic-rich particles are kept suspended in the water column while mineral-rich particles tend to settle out of the water column. Such trapping has been previously described conceptually for other regions, e. g. by Mari et al. (2012) and
30   Ayukai and Wolanski (1997) for river plumes. High $\langle w_s \rangle$ in the transition zone thus implies an enhanced link between pelagic and near-bottom processes. Among them, different transport mechanisms have been discussed that potentially lead, on average, to a net transport of fine sediments shore-wards into the tidal back barriers. *Settling lag* is caused by the different time durations that it takes in different water depths for particles to sediment after the bottom shear stress for erosion falls below the critical one

[revised manuscript text omitted]

**5 Conclusions**

The present study provides  a strong indication of a maximum of sinking velocities along a cross-shore transect that is defined by a gradient of prevailing energy dissipation rates. Towards the off-shore, the enhanced sinking velocities are accompanied by a strong decline of SPMC, and an increasing F/ SPMC ratio. This suggests to define the region as coastal transition zone for  SPMC and composition.

The transition zone is probably an important feature on the course from the coast to the continental shelf.  Predominantly driven by turbulent shear generated by energy dissipation, the transition zone with highest sinking velocities potentially acts as a retention zone for dissolved nutrients leaking from near-shore waters by providing favorable conditions for algae growth. Phytoplankton take up nutrients and excrete extracellular polymeric substances (EPS) that mediate flocculation processes. Embedding minerals into the organic matrix leads to enhanced sinking velocities. Algae thus seem to possess the ability to clear the water column (Fettweis et al., 2014), and link the off-shore dissolved nutrient fluxes to residual landward bottom

fluxes. Consequently, these interlinked processes would have the potential to retain fine sediments and nutrients in coastal areas. The strength of the links eventually affects the eutrophication state of the coastal region.

It is scarcely understood which relevance individual processes have in forming the transition zone.  Besides probably favoring energy dissipation rates of about $\epsilon \approx 10^{-5.5}\,\mathrm{m^2\,s^{-3}}$, algae and
5 their extracellular polymeric excretions are potentially strongly involved as EPS are known to enhance collision efficiency and resistance against fragmentation. Since the primary production in the studied area features a pronounced seasonal cycle, further studies are thus needed to investigate the temporal and spatial extension of the transition zone. Long term or repeated spatial *in situ* measurements, including SPMC, $\epsilon$ profiles and particle properties such as size, $w_s$ and (volume-) composition, are required to underpin the indirectly found enhanced sinking velocities as feature of the transition zone. Additionally, modeling
10 approaches are required to gain a deeper process-based understanding, and to disentangle the closely linked processes among other eutrophication state affecting environmental factors. This also implies the necessity to incorporate biological-mineral interactions in models, especially, when system-wide studies are carried out.

*Author contributions.* R. Riethmüller planed and carried out the ScanFish measurements in the German Bight and took responsibility to ensure data quality. E. M. van der Lee conducted an intensive pre-analysis of the data. U. Gräwe provided hydrodynamical model results
15 from GETM for the times of the cruises. R. Hofmeister handled the 3D GETM results to transfer them to the ScanFish track. J. Maerz and K.W. Wirtz discussed and outlined the general research approach. J. Maerz carried out the data analyses and prepared the manuscript. All co-authors critically contributed to the discussion and to the interpretation of the results and the compilation of the manuscript.

*Acknowledgements.* The authors thank the crew of the research vessel RV Heincke, H. Rink, H. Thomas, M. Heineke, and R. Kopetzky for operating the ScanFish, for processing and quality assurance of the observational data and laboratory work, and L. Merckelbach and
20 G. Flöser for fruitful discussions. We would like to thank the two anonymous reviewers who helped to improve our manuscript. This work was supported *i)* by the German Federal Ministry of Research and Education in the framework of the projects PACE (The future of the Wadden Sea sediment fluxes: still keeping pace with sea level rise?, FKZ 030634A) and MOSSCO (Modular System for Shelves and Coasts, FKZ 03F0667A), *ii)* by the Helmholtz society via the program PACES,  *iii)* by the Niedersächsisches Ministerium für Wissenschaft und Kultur (MWK) and the Niedersächsisches Ministerium für Umwelt und Klimaschutz (MUK) in the framework of the WiMo project
25 (Scientific monitoring concepts for the German Bight), and through the Coastal Observing System for Northern and Arctic Seas (COSYNA). The financing of further developments of the IOW's Baltic Monitoring Program and adaptations of numerical models (STB-MODAT) by the federal state government of Mecklenburg-Vorpommern is greatly acknowledged by Ulf Gräwe. Supercomputing was provided by the North-German Supercomputing Alliance (HLRN). Observational and model data and related material are available for scientific purpose upon request to the corresponding author.

[revised manuscript text omitted]

---

## Author Comment (AC2) · 20 Jun 2016

Dear referee,
thank you very much for your valuable and constructive comments on our manuscript!
Pointing out some underrepresented discussion points, such as about potential further
explanations for spatial heterogeneity of the Wadden Sea eutrophication levels, clearly
helped to improve our manuscript. Your comments are set in italic and respective
answers are set normal

General comments

*The authors analyse an extensive data-set of vertical SPM concentration profiles in the German Bight. They analyse this data-set to reveal the overall cross-shore spatial structure in settling velocities of (aggregates of) particulate matter. They find a maximum in estimated settling velocities in a 15-20m depth zone. The authors put forward two alternative possible explanations for this maximum. First, this could be the zone with optimal balance between turbulence controlled floc-formation and break-up, resulting in the largest flocs and hence the largest settling velocities. Second, they hypothesize that the composition of SPM could vary along a cross-shore gradient. They demonstrate that a functional relation linking settling velocity to primary particle size, floc size, fractal dimension of flocs and relative amount of organic matter in SPM can also show a maximum when all explanatory variables are allowed to vary linearly along a cross-shore gradient. They discuss these observations in terms of the biogeochemical cycling of the German Wadden Sea and hypothesize that along shore differences in the spatial variability of settling velocities could explain regional differences in nutrient cycling and eutrophication status. Although speculative, the discussion of potential implications is interesting. To interpret the observations the authors make use of existing hydrodynamic model simulations, i.c. to estimate the vertical eddy diffusivity and energy dissipation rate. As such they are able to present a 2D data set of SPM concentrations and settling velocities as a 1D relation with energy dissipation rate. This is a nice example of the combined use of in-situ observations and model simulations to retrieve new and insightful information from a system. My major comment is regards the discussion section, which I find short and incomplete. On a first reading it is not clear that the authors are discussing two alternative explanations for the observed maximal settling velocity. This could be made more clear. Neither of both possible explanations of a maximal settling velocity is properly discussed on how realistic it is. No attempt is made to compare the epsilon values at which the a maximum occurs to other research on floc formation and break-up. Is the reported epsilon range of maximal settling ve-*

*locities indeed those that lead to maximal floc size, and under which circumstances? Similar for the results of the conceptual model. Is there any additional support that the different explanatory variables increase linearly over the same distance across the shore? Only the end-members are shortly introduced in relation to other work. It is unclear whether the location of the maximum in the conceptual model can indeed be in the same zone as the observed maximum settling velocity, and under which assumptions. Consequently, it should be discussed which of both potential explanations the authors find the most important, or whether both are deemed equally important. Finally, the authors make no attempt to discuss other explanation for the spatial variability in eutrophication status of the Wadden Sea and why the proposed mechanism here is the most plausible, as seems to be implied by their manuscript. Partially because of the incomplete discussion, I find the conclusions often speculative, e.g.*

We agree that we could have been more explicit about shear rates and related sinking velocities. We therefore further expanded the first paragraph of the discussion by giving more references on shear rates and related maximum sinking velocities.

We now clearly state that our conceptual model is a first order and linear approach to tackle the problem. In the discussion, we now give a brief statement that the location of the transition zone is potentially subject to change when different and non-linear parameter changes are assumed which implies that seasonal seasonal effects can likely happen. The latter requires further investigations.

We tried to make it more clear now that the formation of a sinking velocity maximum is predominantly driven by the apparent shear rate. In addition / on top of that, algae with their excretions potentially modulate the region at which shear rate the maximum sinking velocity occurs. So biological effects likely contribute to the occurrence and extension of the transition zone. Our view is that algae have adapted and rather make use of the general circulation pattern than act against them which complicates the disentangling of the roles of biological and hydrodynamic effects.

We now introduced a whole paragraph about potential further explanations for the heterogeneity of the Wadden Sea eutrophication status. We further tried to make clear that the formation of the transition zone is one among other potential processes leading to the observed differences in the eutrophication status.

*P11 L21-22 "Algae with their seasonality seem to be strongly involved". There is no evidence in the underlying paper supporting this statement. On the contrary, the authors demonstrate that also turbulence controlled flocculation and break-up can explain the existence of a maximal settling velocity. In that case, algae are not involved in the establishment of the coastal transition zone. Overall, I think the authors will be able to address most of the comments with a few additional paragraphs and rewriting the existing text. Therefore I recommend Minor Revisions for this paper.*

We acknowledge the reviewers comment and tried to improve the manuscript in this respect in several places throughout the text. We believe to now better describe the way, how phytoplankton potentially affect the location and establishment of the transition zone in interaction with the given turbulence regime.

Technical points

*The draft needs many textual and technical improvements, and different parts of the methods need further clarification among which: p4 L25 eq 1 Clarify: z-axis pointing*

*downward*

Clarified: "...fluxes in the positively downward pointing vertical direction $z$ ..."

*p4 L28-30. The text motivates why k is taken constant over time (i.e. because settling time scales are much longer than the tidal time scale.). This does not motivate why k can be taken constant over depth, which is the essential assumption to arrive at solution*

[Figure]

*(2). This should be motivated separately. In fact, equation (1) only makes sense as a tidally averaged balance in which $k_v$ is already taken constant in time. Otherwise dC/dt would not vanish, and horizontal advection would not be zero.*

We now state: "... vertically averaged $k_v$ of a profile. This implies the underlying assumption of a rather homogeneous turbulence intensity which we account for in the below described further data processing.", and thus clearly state the underlying assumption. We refer to this assumption again in the motivation for the splitting approach.

*P4 L30 I think the authors are wrong here: the exponential solution of equation (1) assumes zero net fluxes across the vertical boundaries. Otherwise source terms would show up in the solution.*

We stated that we assumed steady state conditions. Thus, the sink and source terms balance each other and are in combination zero. Therefore, source terms can be neglected. See also comment below.

*P4 L30 rephrase: "Assuming zero fluxes across the vertical boundaries and a vertical diffusivity that is constant over depth, equation (1) can be integrated over a water column with arbitrary height $H_p$. This results in an exponential concentration profile. The unknown integration constant can be expressed as a function of the (unknown) average concentration of the profile as: "*

We introduced a subsentence 'allow fluxes across the profiles borders, which cancel out under steady state, we can derive an analytical model' to make clear, why no source and sink terms appear.

*P4 L30: motivate why you express the exponential profile as a function of the unknown average concentration over the profile. You could as well express it as a function of the concentration at a specified depth (e.g. at the top or bottom of the concentration)*

Given the availability of an analytical solution, both methods are equivalent and the depth-averaged observed SPMC is a good estimate for the analytically used mean value.

*P4 L31 $C_m$ is not defined*

We apologize and now defined it. We added $C_m$ after 'analytical model'.

*P4 L31 Be precise in your notation: in this formula $< . >$ denotes a depth average over concentration profile, while later in the text, you state that $< . >$ means an ensemble average.*

We now clearly formulate that it is the vertically averaged $k_v$ value and deleted the word 'ensemble' since it seemed to introduce ambiguity.

*P5 L2-4 It would be helpful to specify the turbulence model that was used in the hydro-dynamic simulations and how vertical turbulent diffusivity was estimated from it.*

We added the sentence: "Vertical mixing is parametrized by means of of a two-equation $k - \epsilon$ turbulence model coupled to an algebraic second-moment closure (Canuto et al. 2001). The implementation of the turbulence module is done via the General Ocean Turbulence Model (GOTM; Umlauf & Burchard 2005)"

*P5 L2 required → requires*

Done.

*P5 L 8 remove comma between "both" and and "observed"*

Done.

*P5 L13-15 How do you motivate the choice for the limits of density gradient standard deviation. You state that they are "somewhat arbitrary", which implicates that they are also "somewhat motivated". Please do so.*

We now write: "Both limits for $\mathrm{std}(\Delta\widetilde{\sigma_T})$ and $\Delta\delta_z\sigma_T$ were applied to select for similar vertically structured observed and modeled density profiles. The chosen values, however, were somewhat arbitrary and therefore considered in a Monte-Carlo-type simulation (see below)"

*P5 L14 See above: here $< . >$ denotes averaging over ensembles. Be precise.*

We deleted 'ensemble' for ambiguity reasons and clarified the meaning for $\sigma'_T$ by introducing: '(here specifically the vertical mean)'

*P5 L14 Clarify: What do you mean by ensemble average, is it the average over an epsilon-bin?*

No, in this context, it is not a bin-wise averaging. We hope that we clarified it by being more explanatory (see our comments above).

*P5 L17-27. I have difficulty to believe this is a correct way to split the vertical SPM profile in subsections. To my best knowledge, density stratification in the German bight is related to water temperature. SPM concentrations and their vertical gradients in the German bight seem too low to me to induce a significant impact on densities and induced turbulence dampening. Large density gradients are in itself the cause of reduced turbulence, whatever the cause of the density gradient. This indeed causes a strongly reduced turbulent diffusivity in the deeper waters compared to the surface waters. As I understand it, this is the principal reason why profiles need to be split in 2 sections on which 2 exponential profiles with constant $k_v$ are fitted, one representative for "deep" waters and the other for "surface" waters . Therefore, the authors should*

*motivate why they not just split the profiles at the level where maximal density gradient exists, and what the difference would be with their method.*

We have introduced a new paragraph to better motivate the chosen procedure. We generally agree that strong density gradients are a good indicator to split the profiles. However, a recent publication by Franks (2014) suggests that density gradients are a rather inadequate proxy for turbulence intensity. We believe that strong vertical SPMC gradients probably indicate vertical turbulence intensity changes. In short, by introducing a splitting dependency on co-occurrence on density and SPMC gradients we thus believe to better acknowledge the turbulence intensity that potentially affect sinking velocity.

*P5 L23. Here again: what kind of averages are denoted with $< . >$. It seems each single profile was considered, thus $< . >$ would mean averaging over depth.*

Yes, it is the depth average. We thus introduced '..., where $\langle X \rangle$ again denotes the vertical average.'

*P5 L34 It took me a while to find what F means. Perhaps this can be clarified here again*

Done.

*P7 L8. Rephrase: "The sensitivity of $w_s$ to changes in each parameter was assessed by varying each parameter separately while keeping the other parameters at their typical values for coastal waters, I.e...."*

Rephrased to: "The sensitivity of $w_s$ to changes in each parameter was assessed by varying each parameter separately while keeping the other parameter parameters at their typical values for coastal waters, i.e. for an organic-rich aggregate with an assumed diameter of $D = 100\,\mu$m, and $\omega = 0.5$, $D_p = 4\,\mu$m, and $d_f = 2$."

*P7 L11 "Vertically and temporally averaged model-derived ..." Over which period was the temporal averaging*

We added the sentence: "The temporal averaging was carried out for the times of the cruises while accounting for the length of the tidal cycle"

*P7 L26 & Fig5 I find the tilted figure a bit difficult to read, why not put it straight up?*

We wanted to emphasize the 3D spatial context, in which the 1D conceptual model is set. To improve readability, we now clearly state what the red and blue lines stand for, which was admittedly not the case before (previously only expressed via the ylabel colors).

*P11 L4 Change "rises" to "raises"*

Done.

Please also note the supplement to this comment:
http://www.biogeosciences-discuss.net/bg-2015-667/bg-2015-667-AC2-supplement.pdf

**Supplement:**

[revised manuscript text omitted]

10    cal solution to observed SPMC profiles  requires information on correspondent $k_{\mathrm{v}}$. These, and additionally energy dissipation rate $\epsilon$ and water density $\sigma_T$ were obtained from hydrodynamical simulations of Gräwe et al. (2015), based on a 1 nautical mile numerical model of the North Sea and the Baltic Sea. In the vertical, 42 terrain-following levels were used. Vertical mixing is parametrized by means of of a two-equation $k - \epsilon$ turbulence model coupled to an algebraic second-moment closure (Canuto et al., 2001). The implementation of the turbulence module is done via the General Ocean Turbulence Model

15    (GOTM; Umlauf and Burchard, 2005). The hydrodynamic parameters were stored as 2 hourly snapshots.

Hydrodynamic model results for $k_v$, $\epsilon$, and $\sigma_T$ were linearly interpolated in space and time to extract a corresponding value for every measured data point. Even though state-of-the-art hydrodynamical models perform generally very well, they may locally exhibit discrepancies to observations. To eventually discriminate for lacking congruency, both  observed and modeled $\sigma_T$, were interpolated on a common vertical grid of $\Delta z = 0.05\,\mathrm{m}$ and filtered by applying a least-square straight

20    line fit to the data and a 'natural' cubic spline interpolant. The latter had a weight of $80\,\%$ to produce a smooth interpolation curve $\sigma_T'(z')$. Subtracting the respective vertical mean resulted in profiles $\widetilde{\sigma_T}$ for both observed and modeled data. We constrained our further analysis to casts satisfying two criteria. First, after subtraction of modeled $\widetilde{\sigma_T}$ from observed data, the standard deviation should not exceed $\mathrm{std}(\Delta\widetilde{\sigma_T}) \leq 0.015\,\mathrm{kg\,m^{-3}}$. Second, the density gradients defined as $\delta_z\sigma_T = (\langle\sigma_T'(\mathrm{last\,meter})\rangle - \langle\sigma_T'(\mathrm{first\,meter})\rangle)/H$, where $\langle x\rangle$ generally represents the  mean of a variable $x$ (here specifically

25    the vertical mean), should not exceed a discrepancy of $\Delta\delta_z\sigma_T \leq 0.015\,\mathrm{kg\,m^{-3}\,m^{-1}}$ between observed and modeled data. Both limits for $\mathrm{std}(\Delta\widetilde{\sigma_T})$ and $\Delta\delta_z\sigma_T$ were applied to select for similar vertically structured observed and modeled density profiles. The chosen values, however, were somewhat arbitrary and therefore considered in a Monte-Carlo-type simulation (see below).

ScanFish cruises covered well-mixed coastal, but partly also stratified waters in the inner German Bight. A consistent approach to retrieve $w_s$ was required to meet both conditions. We chose to split casts into subprofiles, if needed, and to fit

30    the single analytical solution, Eq. (2), to (sub-) profiles. If observed $\delta_z\sigma_T < 5\cdot10^{-4}\,\mathrm{kg\,m^{-3}\,m^{-1}}$, the whole profile was used for fitting, otherwise the cast was split. Since the critical value was set by visual inspection, it was also considered in the Monte-Carlo-type simulation described below. Strong gradients in $\sigma_T$ are typically considered to indicate dampening of

turbulent mixing. The squared buoyancy frequency ($N^2$)

$$N^2 = \frac{g}{\rho} \frac{\partial \rho}{\partial z} \quad , \tag{3}$$

where $g$ denotes the gravitational acceleration constant, is related to the vertical eddy diffusivity $k_v$ by (Osborn and Cox, 1972)

$$k_v = c \frac{\epsilon}{N^2} \tag{4}$$

5     with the current standard value for $c = 0.2$ (Lindborg and Brethouwer, 2008) . A strong vertical sea water density gradient thus reflects low turbulent diffusion, which would imply to split profiles in stratified waters at the maximum density gradient(s). However, as recently discussed by Franks (2014) , mixed layer depths defined by density gradients are a rather inadequate proxy for turbulence intensity. Since particle properties such as size and density are a function of shear rate and SPM components, it is reasonable to expect them to differ with turbulence intensity in a vertical profile leading to different sinking velocities. For

10     example, Leipe et al. (2000) reported vertically varying mean aggregate sizes for the Baltic Sea. While we generally expected vertical gradients in SPMC, strong vertical gradients in SPMC potentially reflect weak mixing and are thus probably an additional indicator for changes in turbulence intensities. The co-occurrence of strong gradients in $\sigma_T$ and SPMC can  thus indicate changes in turbulence intensity and potential particle properties' changes. Splitting at these points allow us to apply the analytical SPMC model (Eq. (2)), where we assumed a rather homogeneous turbulent

15     diffusion by using the vertically averaged $k_v$. Thus, end points of subprofiles were set, where regions in the profile with

$$\left| \frac{1}{C'} \frac{\partial C'}{\partial z'} \cdot \frac{1}{\sigma_T'} \frac{\partial \sigma_T'}{\partial z} \right| > \left\langle \left| \frac{1}{C'} \frac{\partial C'}{\partial z'} \right| \right\rangle \left\langle \left| \frac{1}{\sigma_T'} \frac{\partial \sigma_T'}{\partial z} \right| \right\rangle \tag{5}$$

occur, and reach their maximum, where $\langle x \rangle$ again denotes the vertical average. $C'$ is the SPMC smoothing spline-interpolated analogously to $\sigma_T$. Start points of the subprofiles were either the first data point from surface or where regions defined by Eq. (5) end with increasing depth. Only subprofiles longer than $4\,\mathrm{m}$ were considered in the further analysis.

[revised manuscript text omitted]

5 (according to Maggi, 2009) and $\rho_o = 1100\,\text{kg}\,\text{m}^{-3}$ (as an average value of the range given in Fettweis, 2008). The water density and the dynamic viscosity are set to $\rho = 1000\,\text{kg}\,\text{m}^{-3}$ and $\mu = 0.001\,\text{kg}\,\text{m}^{-1}\,\text{s}^{-1}$, respectively.  The sensitivity of $w_s$ to changes in each parameter was assessed by varying each parameter separately while keeping the other parameter parameters at their typical values for coastal waters, i.e. for an organic-rich aggregate with an assumed diameter of $D = 100\,\mu\text{m}$, and $\omega = 0.5$,

10 $D_p = 4\,\mu\text{m}$, and $d_f = 2$.

**3 Results**

**3.1 Cross-coastal gradients**

Energy dissipation rates $\epsilon$ generally possess spatio-temporal variability. Most relevant for this study, vertically and temporally averaged model-derived $\epsilon$ exhibit a cross-shore gradient from the inner German Bight towards the coast. The temporal

15 averaging was carried out for the times of the cruises while accounting for the length of the tidal cycle. Values range between $\epsilon \approx 10^{-9}\,\text{m}^2\,\text{s}^{-3}$ and $\epsilon \approx 10^{-4}\,\text{m}^2\,\text{s}^{-3}$, respectively (Fig. 3).

Along this range, mean SPMC first increased from values below $1\,\text{g}\,\text{m}^{-3}$ to approximately $10\,\text{g}\,\text{m}^{-3}$. Above $\log_{10}(\epsilon) = -5.5$, however, mean SPMC moderately decreased to approximately $6\,\text{g}\,\text{m}^{-3}$ and increased again to $10\,\text{g}\,\text{m}^{-3}$ with further increasing $\epsilon$ (Fig. 4 A) towards the coast.

20  Under the assumption that fluorescence can be applied as a proxy for POM, the potential significance of algae and their products for particle composition is depicted by the mean ratio between measured fluorescence and SPMC.  High F/SPMC ratios likely indicate rather organic-rich particles, while low F/SPMC ratios potentially indicate rather mineral particle-dominated flocs. The F/SPMC ratio shows rather high variability for $\log_{10}(\epsilon) < -6$ (Fig. 4 B). With further increasing $\epsilon$, the ratio drops and levels at approximately a fourth of the open German Bight value for $\log_{10}(\epsilon) > -5$.

25 Average sinking velocities $\langle w_s \rangle$ show very low values of order $10^{-6}\,\text{m}\,\text{s}^{-1}$ to $10^{-5}\,\text{m}\,\text{s}^{-1}$ in regions of $\log_{10}(\epsilon) < -7.5$ (Fig. 4 C). At higher $\epsilon$, $\langle w_s \rangle$ increased significantly reaching a maximum of about $7 \cdot 10^{-4}\,\text{m}\,\text{s}^{-1}$ around $\log_{10}(\epsilon) \approx -5.5$. This maximum $\langle w_s \rangle$ around $\log_{10}(\epsilon) \approx -5.5$ coincided with the increase of SPMC, and the drop in fluorescence to SPMC ratio. With further increasing $\epsilon$, $\langle w_s \rangle$ decreased again to values of $\langle w_s \rangle \approx 3 \cdot 10^{-4}\,\text{m}\,\text{s}^{-1}$. The Monte-Carlo-type simulation to estimate parameter uncertainties exhibits the same pattern, expressed as $\sigma$ and $2\sigma$ confidence levels, $68\,\%$ and $95\,\%$ respectively

30 (Fig. 4C), around their ensemble mean, and thus underpins the results of $\langle w_s \rangle$ along $\epsilon$.

**3.2 Conceptual model**

 Sinking velocity of an average particle (as parametrized in Sec. 2.4) changes, after variation in single parameters along the transect from open to coastal waters, in an almost linear way (Fig. 5, upper four panels). Flocs would sink less rapid because of decreasing diameter of the aggregate or the primary particles in the coastal region. By contrast, increasing amount of  mineral particles and fractal dimension would both lead to increased $w_s$. As an overall effect, when considering all parameters, $w_s$ reaches a maximum in between the near-shore and open waters (lowest panel of Fig. 5).

**4 Discussion**

**4.1 Sinking velocity on a gradient of prevailing energy dissipation rate**

Sinking velocities have been determined along a cross-shore transect in the German Bight defined by the prevailing $\epsilon$. For the reconstruction of $\langle w_s \rangle$, we had to assume congruency between *in situ* measurements and hydrodynamic model results within a range defined by the applied filters. The estimated $\langle w_s \rangle$ of order $10^{-6}\,\mathrm{m\,s^{-1}}$ to $5 \cdot 10^{-4}\,\mathrm{m\,s^{-1}}$ in low and high energy dissipation regions, respectively (Fig. 4), compare very well with former *in situ* studies in the German Bight and adjacent areas. Puls et al. (1995) found $w_s \approx 10^{-6}\,\mathrm{m\,s^{-1}}$ to $2 \cdot 10^{-5}\,\mathrm{m\,s^{-1}}$ for the open German Bight, a low $\epsilon$ region, and Pejrup and Mikkelsen (2010) reported median $w_s$ of $1 \cdot 10^{-4}\,\mathrm{m\,s^{-1}}$ to $11 \cdot 10^{-4}\,\mathrm{m\,s^{-1}}$ for the Rømø and Høyer Dyb, Danish Wadden Sea where $\epsilon$ is comparably high. Our estimated $\langle w_s \rangle$ are also well in the range found in other regions, e. g. in Chesapeake Bay (Gibbs, 1985) or at the Belgian Coast (Fettweis and Baeye, 2015). In agreement with former studies (e. g. compiled in Dyer, 1989), obtained $\langle w_s \rangle$ increase with increasing $\langle \mathrm{SPMC} \rangle$ (Fig. 6), which gave further confidence in the methodological approach. The correlation is, however, rather poor compared to former local studies and can be explained by the heterogeneity of the German Bight system, seasonal effects and the intrinsically high variability of sinking velocities (Fettweis, 2008) and of course of turbulence (Pejrup and Mikkelsen, 2010). With respect to prevailing $\epsilon$, the significant maximum of $\langle w_s \rangle$ at $\langle \epsilon \rangle \approx 10^{-5.5}\,\mathrm{m^2\,s^{-3}}$ can be explained by the balancing between aggregation and fragmentation. Both processes are controlled by turbulent shear, generated by energy dissipation, SPM volume concentration, and adhesion properties of particles involved. Previous flocculation modeling studies pointed to a formation of a $w_s$ maximum along $\epsilon$ (Winterwerp, 1998; Pejrup and Mikkelsen, 2010). Our findings of a maximum in $\langle w_s \rangle$ between $\epsilon \approx 10^{-6}\,\mathrm{m^2\,s^{-3}}$ and $10^{-5}\,\mathrm{m^2\,s^{-3}}$  which translates to shear rates of about $1\,\mathrm{s^{-1}}$ to about $3.2\,\mathrm{s^{-1}}$ for $\nu = 10^{-6}\,\mathrm{m^2\,s^{-1}}$, should thus be comparable to former theoretical studies. For example, in a field work-based modeling approach Pejrup and Mikkelsen (2010) found the maximum of sinking velocity to occur under higher shear rates of about $8.5\,\mathrm{s^{-1}}$. By contrast, Maerz et al. (2011) calculated highest mean sinking velocities of about $5 \cdot 10^{-4}\,\mathrm{m\,s^{-1}}$, similar to our findings, for a jar test device-based lab experiment with natural SPM during shear rates of about $1\,\mathrm{s^{-1}}$. Given the underlying uncertainties in calculating the shear rate in both cases, our study thus agrees well with former theoretical studies. Hence, as already suggested by Dyer (1989) and Pejrup and Mikkelsen (2010), in addition to SPMC, turbulent shear can be regarded as the major determinant for $w_s$.

**4.2 Sinking velocity as a result of SPMC, composition and a turbulence gradient**

Our study additionally suggests a change in particle composition with $\epsilon$ as proxied by the fluorescence to SPMC ratio. With $\epsilon$, SPMC increases towards the coast and mineral fraction becomes dominant compared to organic particles at low $\epsilon$ (Eisma and Kalf, 1987). Hence, density and likely other physico-chemical properties of primary particles such as size and
5    adhesion change accordingly and thus influence $w_s$ along a cross-shore transect. Our conceptual model illustrates the effect of varying particle properties on $w_s$ independent of the observations used in our data analysis. However, the course of the parameters implicitly presumes certain environmental conditions, as e. g. size of particles is, among others, a function of SPMC and shear. Single parameter changes in the conceptual model resulted in mostly linear responses in $w_s$, while considering all parameters led to a maximum in $w_s$ on the conceptual cross-shore transect, in alignment with our findings in the data analysis.
10   The assumption of linearly changing parameters is a first order approximation. Differently and non-linear changing parameters would lead to a transformation of the maximum sinking velocity zone such as a shifting and / or stretching (not shown). Former model studies have described for a given shear regime how varying particle properties such as particle adhesion lead to changing particle sizes and thus sinking velocity (e. g. Maerz et al., 2011) . Hence, higher particle adhesion would translate to larger particles and stronger fragmentation resistance which would allow particles sink out of the water column closer to
15   the coast under higher turbulence intensities. This implies that spatial changes in SPMC and composition, and accordingly physico-chemical properties of particles, potentially affect the location of the transition zone. As discussed below this probably happens on a seasonal time scale due to the modulation of the transition zone by phytoplankton exudations which requires further investigations. While the cross-shore distribution of $w_s$ is predominantly controlled by prevailing shear, the conceptual model indicates  a potential additional relevance of physico-chemical properties for the formation of the high sinking
20   velocity zone.

**4.3 Implications of a cross-coastal maximum of sinking velocity for biogeochemical cycling in the coastal zone**

The region of $\log_{10}(\epsilon) \approx -5.5$ with highest $\langle w_s \rangle$ coincides with a zone located off the coast at depth of about $15\,\mathrm{m}$ to $20\,\mathrm{m}$. It is accompanied by a strong gradient in  SPMC and likely composition as depicted by the F/SPMC ratio. This  suggests that the region can be considered as a transition zone, hindering mineral particles to escape further off-
25   shore. To simplify, consider the course from the coast to open waters. Once formed, relatively dense, fast sinking flocs, whose properties are adapted to the transition zones' turbulence level, would easily settle out of the water column once transported offshore. That is because turbulence becomes too weak to break those aggregates apart and to retain them in suspension. Thus, only loose, organic-rich particles are kept suspended in the water column while mineral-rich particles tend to settle out of the water column. Such trapping has been previously described conceptually for other regions, e. g. by Mari et al. (2012) and
30   Ayukai and Wolanski (1997) for river plumes. High $\langle w_s \rangle$ in the transition zone thus implies an enhanced link between pelagic and near-bottom processes. Among them, different transport mechanisms have been discussed that potentially lead, on average, to a net transport of fine sediments shore-wards into the tidal back barriers. *Settling lag* is caused by the different time durations that it takes in different water depths for particles to sediment after the bottom shear stress for erosion falls below the critical one

[revised manuscript text omitted]

**5 Conclusions**

The present study provides  a strong indication of a maximum of sinking velocities along a cross-shore transect that is defined by a gradient of prevailing energy dissipation rates. Towards the off-shore, the enhanced sinking velocities are accompanied by a strong decline of SPMC, and an increasing F/ SPMC ratio. This suggests to define the region as coastal transition zone for  SPMC and composition.

The transition zone is probably an important feature on the course from the coast to the continental shelf.  Predominantly driven by turbulent shear generated by energy dissipation, the transition zone with highest sinking velocities potentially acts as a retention zone for dissolved nutrients leaking from near-shore waters by providing favorable conditions for algae growth. Phytoplankton take up nutrients and excrete extracellular polymeric substances (EPS) that mediate flocculation processes. Embedding minerals into the organic matrix leads to enhanced sinking velocities. Algae thus seem to possess the ability to clear the water column (Fettweis et al., 2014), and link the off-shore dissolved nutrient fluxes to residual landward bottom

fluxes. Consequently, these interlinked processes would have the potential to retain fine sediments and nutrients in coastal areas. The strength of the links eventually affects the eutrophication state of the coastal region.

It is scarcely understood which relevance individual processes have in forming the transition zone.  Besides probably favoring energy dissipation rates of about $\epsilon \approx 10^{-5.5}\,\mathrm{m^2\,s^{-3}}$, algae and
5 their extracellular polymeric excretions are potentially strongly involved as EPS are known to enhance collision efficiency and resistance against fragmentation. Since the primary production in the studied area features a pronounced seasonal cycle, further studies are thus needed to investigate the temporal and spatial extension of the transition zone. Long term or repeated spatial *in situ* measurements, including SPMC, $\epsilon$ profiles and particle properties such as size, $w_s$ and (volume-) composition, are required to underpin the indirectly found enhanced sinking velocities as feature of the transition zone. Additionally, modeling
10 approaches are required to gain a deeper process-based understanding, and to disentangle the closely linked processes among other eutrophication state affecting environmental factors. This also implies the necessity to incorporate biological-mineral interactions in models, especially, when system-wide studies are carried out.

*Author contributions.* R. Riethmüller planed and carried out the ScanFish measurements in the German Bight and took responsibility to ensure data quality. E. M. van der Lee conducted an intensive pre-analysis of the data. U. Gräwe provided hydrodynamical model results
15 from GETM for the times of the cruises. R. Hofmeister handled the 3D GETM results to transfer them to the ScanFish track. J. Maerz and K.W. Wirtz discussed and outlined the general research approach. J. Maerz carried out the data analyses and prepared the manuscript. All co-authors critically contributed to the discussion and to the interpretation of the results and the compilation of the manuscript.

*Acknowledgements.* The authors thank the crew of the research vessel RV Heincke, H. Rink, H. Thomas, M. Heineke, and R. Kopetzky for operating the ScanFish, for processing and quality assurance of the observational data and laboratory work, and L. Merckelbach and
20 G. Flöser for fruitful discussions. We would like to thank the two anonymous reviewers who helped to improve our manuscript. This work was supported *i)* by the German Federal Ministry of Research and Education in the framework of the projects PACE (The future of the Wadden Sea sediment fluxes: still keeping pace with sea level rise?, FKZ 030634A) and MOSSCO (Modular System for Shelves and Coasts, FKZ 03F0667A), *ii)* by the Helmholtz society via the program PACES,  *iii)* by the Niedersächsisches Ministerium für Wissenschaft und Kultur (MWK) and the Niedersächsisches Ministerium für Umwelt und Klimaschutz (MUK) in the framework of the WiMo project
25 (Scientific monitoring concepts for the German Bight), and through the Coastal Observing System for Northern and Arctic Seas (COSYNA). The financing of further developments of the IOW's Baltic Monitoring Program and adaptations of numerical models (STB-MODAT) by the federal state government of Mecklenburg-Vorpommern is greatly acknowledged by Ulf Gräwe. Supercomputing was provided by the North-German Supercomputing Alliance (HLRN). Observational and model data and related material are available for scientific purpose upon request to the corresponding author.

[revised manuscript text omitted]

---

## Author Response (AR1)

Dear Mr. Juniper,

thank you very much for accompanying the process of the manuscript submission and revision. According to your comment, we carefully revised the manuscript with respect to English grammar and polished the style if required. A marked-up version of the manuscript is attached below. Please find changes marked in red and the corrections marked in blue.

Sincerely yours, in behalf of all co-authors,

J. Maerz

**Maximum sinking velocities of suspended particulate matter in a coastal transition zone**

Joeran Maerz[1], Richard Hofmeister[1], Eefke M. van der Lee[1], Ulf Gräwe[2,3], Rolf Riethmüller[1], and Kai W. Wirtz[1]

[1]Institute of Coastal Research, Helmholtz-Zentrum Geesthacht (HZG), Geesthacht, Germany
[2]Leibniz Institute for Baltic Sea Research, Warnemünde, Germany
[3]Institute of Meteorology and Climatology, Leibniz University Hannover, Hannover, Germany

*Correspondence to:* Joeran Maerz (present address: joeran.maerz@mpimet.mpg.de)

**Abstract.** Marine coastal ecosystem functioning is crucially linked to the transport and fate of suspended particulate matter (SPM). Transport of SPM is, amongst others, controlled by sinking velocity $w_s$. Since $w_s$ of cohesive SPM aggregates varies significantly with size and composition of mineral and organic origin, $w_s$  exhibits large spatial variability along gradients of turbulence, SPM concentration (SPMC) and SPM composition. In this study, we retrieved $w_s$ for the German Bight, North Sea, by combining measured vertical turbidity profiles with simulation results for turbulent eddy diffusivity.  We analyzed $w_s$ with respect to modeled prevailing  dissipation rates $\epsilon$  and found that mean $w_s$ were significantly enhanced around $\log_{10}(\epsilon\,(\mathrm{m^2\,s^{-3}})) \approx -5.5$. This $\epsilon$ region is typically found at water depths of approximately $15\,\mathrm{m}$ to $20\,\mathrm{m}$  along cross-shore  transects. Across this zone, SPMC declines drastically towards the offshore and a change in particle composition occurs. This characterizes a transition zone with potentially enhanced vertical fluxes. Our findings contribute to the conceptual understanding of nutrient cycling in the coastal region which is as follows: Previous studies identified an estuarine circulation. Its residual landward-oriented bottom currents are  loaded with SPM, particularly within the transition zone. This retains and traps fine sediments and particulate-bound nutrients in coastal waters where organic components of SPM become re-mineralized. Residual surface currents transport dissolved nutrients  off shore, where they are again consumed by phytoplankton. Algae excrete extracellular polymeric substances which are known to mediate mineral aggregation and thus sedimentation. This probably takes place particularly in the transition zone and completes the coastal nutrient cycle. The efficiency of the transition zone for retention is thus suggested as an important mechanism that underlies the often observed nutrient gradients towards the coast.

**1 Introduction**

Biogeochemical cycling and functioning of marine coastal and shelf sea systems crucially relies on particle transport. Vertical fluxes of suspended particulate matter (SPM) are determined by sinking velocity $w_s$ and indirectly affect the horizontal transport. In coastal systems, SPM is composed of living and non-living particulate organic matter (POM) and fine cohesive and non-cohesive resuspended minerals. Fine-grained minerals of sizes typically up to $8\,\mu\mathrm{m}$ (Chang et al., 2006) and POM can undergo aggregation and fragmentation processes that change sinking velocity and thus transport properties. As a consequence

of flocculation, SPM aggregates ubiquitously possess a broad spectrum of size and composition (Fettweis, 2008). This hetero-geneity  between flocs increases the methodological effort to analyze $w_s$ *in situ* (Fettweis, 2008). On larger scales, SPM concentration (SPMC) and composition additionally exhibit strong spatio-temporal variability that are the result of manifold interplaying processes. Tidal and wind-induced currents are the major driver for resuspension and subsequent horizontal

5   transport, while biological processes,  such as algae growth and bio-induced sediment stabilization (Black et al., 2002; Stal, 2003), interfere and thus  shape the complex distribution of SPM in coastal and estuarine systems. Typically, SPMC and composition show  cross-shore gradients (Tian et al., 2009; van der Lee et al., 2009; Li et al., 2010). In shallow waters near the coast, where turbulence and thus resuspension  is high, SPMC consists of flocs that are mainly composed of mineral particles with high densities. By contrast, in deeper off-shore regions, SPMC is lower and the flocs are

10   looser and more organic. This  general pattern of changing SPMC and composition from coast to open waters is  typical for our research area, the German Bight (Eisma and Kalf, 1987),  and is observed worldwide in estuaries (e. g. Fugate and Friedrichs, 2003) and across coastal seas (e. g. van der Lee et al., 2009). Since both  SPMC and turbulence  control $w_s$ of cohesive material (Pejrup and Mikkelsen, 2010), it is likely that these  cross-shore SPM gradients induce considerable spatial variability  in $w_s$ and thus affect transport and fate of SPM in coastal marine systems.  However,

15   to date and to the authors' knowledge,  no comprehensive analysis has addressed system-wide cross-shore gradients in sinking velocity, especially in relation to possible drivers.

    Sinking velocity of SPM is determined by floc size and density, both resulting from a complex interplay of processes. Particle size distribution is locally governed by restructuring (Becker et al., 2009), aggregation and processes mainly driven by turbulence-induced shear, $\overline{G} = (\epsilon/\nu)^{1/2}$ , (Pejrup and Mikkelsen, 2010) . This turbulence is

20   generated by energy dissipation $\epsilon$ (Camp and Stein, 1943) with kinematic viscosity $\nu$  . Under a given shear regime, aggregation is controlled by floc volume concentration  and adhesion properties of the  primary particles of mineral and organic origin that form the floc. The higher the volume concentration of flocs, the higher is the encounter rate. Subsequently, adhesion forces of the involved particles eventually determine the probability to stick together. In addition, adhesive forces limit the intrusion of particles during clustering  and loosens the floc structure

25   (Meakin and Jullien, 1988). This leads to a floc morphology that shows self-similar fractal scaling (Kranenburg, 1994). As a result, aggregates possess decreased density compared to the comprising primary particles.  In addition, adhesion forces between the particles within a floc  strengthen the resistance of aggregates against fragmentation (Kranenburg, 1999), while the smallest eddies, with sizes of the Kolmogorov microscale (Kolmogorov, 1941), potentially limit the maximum particle size (Berhane et al., 1997). In sum, $w_s$ is locally governed by turbulence-driven processes whose rates depend on the

30   physico-chemical properties and volume concentration of particles.

    To date, there is still a lack of understanding of how environmental conditions and especially biological processes affect physico-chemical properties, and thus $w_s$  of cohesive SPM. Typically, a power law relation between median $w_s$ and SPMC is postulated and found at local measurements (Dyer, 1989).  However, these relations vary considerably in their power factor among different systems as  summarized by Dyer (1989) for various estuaries. This variation

35   can be attributed to different shear stresses and physico-chemical properties of involved particles. They are particularly

subjected to algae and microbial extracellular polymeric substances (EPS) excretions  such as transparent exopolymer particles (TEP)  that are known to mediate aggregation processes. TEP bridge and glue mineral particles together (Decho, 1990)  and potentially increase resistance against particle fragmentation (Fettweis et al., 2014). By these mechanisms, TEP are hypothesized to enable phytoplankton to clear the water column from suspended sediments (Fettweis et al., 2014).

5 Such clearing ability may contribute to spatial variability of $w_s$ which would affect biogeochemical cycles. So far, knowledge on spatial variations of $w_s$ on a system scale is rare. A study on a transect in San Francisco Bay  was carried out (Manning and Schoellhamer, 2013), but generally, there is no systematic understanding about the relevance of sedimentation variability for biogeochemical cycles in coastal ecosystems. Such knowledge, though, is needed to understand and model coastal shelf biogeochemistry and sedimentology. For example, it is still an ongoing scientific discussion which processes

10 and to what extent  do they sustain the net-sediment transport towards the Dutch, German and Danish coast into the Wadden Sea (Postma, 1984). We therefore  aim at a reconstruction and analyses of $w_s$ of SPM for the German Bight, North Sea. We  develop a new approach to retrieve $w_s$ from high resolution turbidity profiles in combination with vertical mixing rates estimated by a hydrodynamical model. Our findings are particularly discussed in  light of their relevance for biogeochemical cycling of matter in coastal waters.

15 ## 2 Methods

In the following section, the study area, sampling and preprocessing of observational data are described.  We further explain the procedure for extracting sinking velocities from observations with the aid of hydrodynamical model results .

**2.1 Study area**

20 The surveyed German Bight (Fig. 1) is located in the south-east of the North Sea and features a typical depth of about $30\,\mathrm{m}$ to  a maximum of $50\,\mathrm{m}$. The North Sea is a shallow shelf sea with an average depth of $80\,\mathrm{m}$ (Sündermann and Pohlmann, 2011) and is connected to the North Atlantic via the English Channel in the south-west  and opens towards the north , spanning the European continental shelf. The North Sea is exposed to tides whose tidal range is between approximately $1.8\,\mathrm{m}$ and $3.4\,\mathrm{m}$ in the southeastern German Bight. The main North Sea tidal wave  turns anti-clockwise and

25 drives the large scale current system (Sündermann and Pohlmann, 2011). SPM originating from the British coast is transported eastwards by the East Anglian Plume and occasionally reaches the surveyed area (Fettweis et al., 2012; Pietrzak et al., 2011). In the southern North Sea, SPM and dissolved nutrients that  enters though the river Rhine are transported eastward along the Dutch and German East-Frisian shore by this  large-scale current system. In addition, the German rivers Ems, Weser and Elbe discharge into the German Bight. As a result of the riverine nutrient input, the German Bight features a

[revised manuscript text omitted]

This describes the balance between fluxes in the positively downward pointing vertical direction $z$ due to sinking and turbulent eddy diffusivity $k_{\mathrm{v}}$. If we assume that the sinking time scale is larger than the tidal period, we can simplify Eq. (1) by using the vertically averaged $k_v$ of a profile. This assumes a rather homogeneous turbulence intensity which we account for in the data processing described below. If we further allow fluxes across profiles borders, which cancel out under steady state, we can derive an analytical model ($C_{\mathrm{m}}(z)$) for depth-dependent SPMC

$$C_{\mathrm{m}}(z) = \frac{\lambda \exp(\lambda z^*/H_{\mathrm{p}})}{\exp(\lambda) - 1} \langle C \rangle \quad \text{where } \lambda = \frac{w_s H_{\mathrm{p}}}{\langle k_v \rangle}. \tag{2}$$

Here, $H_{\mathrm{p}}$ represents the profile height and $z^* = z - z_1$, where $z_1$ is the depth at which the profile starts. Application of the analytical solution to observed SPMC profiles requires information on corresponding $k_{\mathrm{v}}$ values. These and energy dissipation rate $\epsilon$ and water density $\sigma_T$ were obtained from hydrodynamical simulations of Gräwe et al. (2015), based on a 1 nautical mile numerical model of the North Sea and the Baltic Sea. In the vertical, 42 terrain-following levels were used. Vertical mixing was parametrized by means of of a two-equation $k - \epsilon$ turbulence model coupled to an algebraic second-moment closure (Canuto et al., 2001). The implementation of the turbulence module was done via the General Ocean Turbulence Model (GOTM; Umlauf and Burchard, 2005). The hydrodynamic parameters were stored as 2-hourly snapshots.

Hydrodynamic model results for $k_v$, $\epsilon$, and $\sigma_T$ were linearly interpolated in space and time to extract a corresponding value for every measured data point. Even though state-of-the-art hydrodynamical models generally perform very well, they may locally exhibit discrepancies with observations. To discriminate the lack of congruency, both observed and modeled $\sigma_T$ were interpolated on a common vertical grid of $\Delta z = 0.05\,\mathrm{m}$ and filtered by applying a least-square straight line fit to the data and a 'natural' cubic spline interpolant. The latter had a weight of $80\,\%$ to produce a smooth interpolation curve $\sigma'_T(z')$. Subtracting the respective vertical mean resulted in profiles $\widetilde{\sigma_T}$ for both observed and modeled data. We further constrained our analysis to casts that satisfy two criteria. First, after subtraction of modeled $\widetilde{\sigma_T}$ from observed data, the standard deviation should not exceed $\mathrm{std}(\Delta\widetilde{\sigma_T}) \leq 0.015\,\mathrm{kg}\,\mathrm{m}^{-3}$. Second, the density gradients defined as $\delta_z\sigma_T = (\langle \sigma'_T(\mathrm{last\,meter}) \rangle - \langle \sigma'_T(\mathrm{first\,meter}) \rangle)/H$, where $\langle x \rangle$ represents the mean of a variable $x$ and specifically in this case, the vertical mean, should not exceed a difference of $\Delta\delta_z\sigma_T \leq 0.015\,\mathrm{kg}\,\mathrm{m}^{-3}\,\mathrm{m}^{-1}$ between observed and modeled data. Both limits for $\mathrm{std}(\Delta\widetilde{\sigma_T})$ and $\Delta\delta_z\sigma_T$ were applied to select similar vertically structured observed and modeled density profiles. The chosen values, however, were somewhat arbitrary and therefore considered in a Monte-Carlo-type sensitivity simulation (see below).

ScanFish cruises covered well-mixed coastal but partly also stratified waters in the inner German Bight. A consistent approach to retrieve $w_s$ was required to meet both conditions. When needed, we chose to split casts into subprofiles, if

 and to fit the single analytical solution, Eq. (2), to (sub-) profiles. If observed $\delta_z \sigma_T < 5 \cdot 10^{-4}\,\mathrm{kg\,m^{-3}\,m^{-1}}$, the whole profile was used for fitting, otherwise the cast was split. Since the critical value was set by visual inspection,  sensitivity of this critical value was also considered in the Monte-Carlo-type simulation described below. Strong gradients in $\sigma_T$ are typically considered to indicate dampening of turbulent mixing. The squared buoyancy frequency ($N^2$)

$$N^2 = \frac{g}{\rho}\frac{\partial \rho}{\partial z} \quad , \tag{3}$$

where $g$ denotes the gravitational acceleration constant, is related to the vertical eddy diffusivity $k_v$ by (Osborn and Cox, 1972)

$$k_v = c\,\frac{\epsilon}{N^2} \tag{4}$$

with the current standard value for $c = 0.2$ (Lindborg and Brethouwer, 2008). A strong vertical sea water density gradient thus reflects low turbulent diffusion, which would imply to split profiles in stratified waters at the maximum density gradient(s). However, as recently discussed by Franks (2014), mixed layer depths defined by density gradients are a rather inadequate proxy for turbulence intensity. Since particle properties such as size and density are a function of shear rate and SPM components, it is reasonable to expect them to differ with  vertical turbulence intensity, thus leading to different sinking velocities. For example, Leipe et al. (2000) reported vertically varying mean aggregate sizes for the Baltic Sea. While we generally  expect vertical gradients in SPMC, strong vertical gradients in SPMC potentially reflect weak mixing and are thus  possible indication for changes in turbulence intensities. The co-occurrence of strong gradients in $\sigma_T$ and SPMC can thus  signal changes in turbulence intensity and  particle property. Splitting at these points  allowed us to apply the analytical SPMC model (Eq. (2)), where we assumed a rather homogeneous turbulent diffusion by using the vertically averaged $k_v$. Thus, end points of subprofiles were set, where regions in the profile with

$$\left|\frac{1}{C'}\frac{\partial C'}{\partial z'} \cdot \frac{1}{\sigma_T'}\frac{\partial \sigma_T'}{\partial z}\right| > \left\langle\left|\frac{1}{C'}\frac{\partial C'}{\partial z'}\right|\right\rangle \left\langle\left|\frac{1}{\sigma_T'}\frac{\partial \sigma_T'}{\partial z}\right|\right\rangle \tag{5}$$

occur  and reach their maximum. Here, $\langle x \rangle$  denotes the vertical average  and $C'$ is the SPMC smoothing spline-interpolated analogously to $\sigma_T$. Start points of the subprofiles were either the first data point from surface or where regions defined by Eq. (5) end with increasing depth. Only subprofiles longer than $4\,\mathrm{m}$ were considered  for further analysis.

The analytical model, Eq. (2), was fitted to the original observation of each (sub-) profile with $N_p$ data points, and the cost function

$$\mathrm{err} = \frac{1}{N_p} \sum_{i=1}^{N_c} \frac{(C(z_i) - C_\mathrm{m}(z_i))^2}{0.5\,(\delta_R C^2 + \delta_P C^2)} \tag{6}$$

was calculated accordingly. $\delta_P C^2$ and $\delta_R C^2$ represent the variance  in concentration of a profile and in a region, respectively. The latter  was defined as the variance for measurements around the profile within $\pm 5$ min. to account for higher  variability in coastal than in open water regions found in a pre-analysis of the data. Only profiles below a cost function value of 0.05, chosen by visual inspection, were considered for the analysis. The splitting and subsequent fitting of the analytical

model is visualized in Fig. 2. The variables SPMC  and fluorescence-to-SPMC ratio (F/SPMC) were vertically averaged for the respective (sub-) profiles. Afterward, variables were binned with respect to modeled $\epsilon$ and eventually averaged bin-wise, resulting in  respective bin-wise ensemble means $\langle\text{SPMC}\rangle$, $\langle\text{F/SPMC}\rangle$, $\langle w_s\rangle$, and $\langle\epsilon\rangle$. To test for significant changes of binned $\langle w_s\rangle$ with $\langle\epsilon\rangle$, we applied a  Mann-Whitney-U test with a significance level of $p < 0.05$ for each binned $\langle w_s\rangle$ against each other. In summary, approximately 67 % of initially measured  $\approx 68000$ profiles passed the congruency check with modeled ones. After application of the cost function threshold, this resulted in  about 12260 $w_s$-values.

All applied threshold values were carefully selected by visual inspection during each filtering step. To quantify their influence on the result, we performed a Monte-Carlo-type simulation with a variation by $\pm 50\,\%$ for the following parameters: $\text{std}(\Delta\widetilde{\sigma_T})$, $\Delta\delta_z\sigma_T$, $\delta_z\sigma_T$, and the cost function error. The analysis was repeated for variations of all parameters against each other. For each parameter variation, binned mean values for $\langle w_s\rangle$ versus $\epsilon$ were calculated, eventually averaged and their standard deviation $\sigma$ quantified (see Fig. 4 C).

**2.4 Conceptual cross-coastal sinking velocity model**

We introduce and apply a conceptual model for the cross-shore variation of $w_s$ as a tool to interpret our results. According to Stokes (1851), $w_\text{s}$ of a particle of diameter $D$ can be described by

$$w_\text{s} = \frac{1}{18\,\mu}\left(\rho_\text{f} - \rho\right)g\,D^2 \quad , \tag{7}$$

where $\mu$ is the dynamic viscosity, $\rho$ is the water density, and $g$ the gravitational acceleration constant. The floc density $\rho_\text{f}$ is strongly related to the particles' composition and structure. The latter can be described as fractal dimension $d_\text{f}$ according to  Kranenburg (1994), who derived the excess floc density $\Delta\rho_\text{f} = \rho_\text{f} - \rho$ for a particle composed of primary particles of diameter $D_\text{p}$ and density $\rho_\text{p}$:

$$\Delta\rho_\text{f} = (\rho_\text{p} - \rho)\left(\frac{D_\text{p}}{D}\right)^{3-d_\text{f}} \quad . \tag{8}$$

Following the approach of Kranenburg (1994) and assuming equally sized primary particles of diameter $D_\text{p}$, Maggi (2009) derived the excess floc density for an aggregate composed of mineral and organic particles with density $\rho_\text{s}$ and $\rho_\text{o}$, respectively,

$$\Delta\rho_\text{f} = (\omega\Delta\rho_\text{s} + (1-\omega)\Delta\rho_o)\cdot\left(\frac{D_\text{p}}{D}\right)^{3-d_\text{f}} \quad , \tag{9}$$

 For $w_s$ of an aggregate, it follows

$$w_\text{s} = \frac{1}{18\,\mu}(\omega\Delta\rho_\text{s} + (1-\omega)\Delta\rho_o)\,g\,D_\text{p}^{3-d_\text{f}}D^{d_\text{f}-1} \quad , \tag{10}$$

where $\omega = n_\text{s}/(n_\text{s} + n_\text{o})$ and $n_\text{s}$ and $n_\text{o}$ are the number of mineral and organic

$$w_\text{s} = \frac{1}{18\,\mu}(\omega\Delta\rho_\text{s} + (1-\omega)\Delta\rho_o)\,g\,D_\text{p}^{3-d_\text{f}}D^{d_\text{f}-1} \quad .$$

primary particles, respectively.

5    This simple $w_s$ model, Eq. (10), allows for calculation of trends in $w_s$ on a cross-shore transect. Based on observations, we  presume the general following parameter changes from coastal to open waters: *i)* average primary particle size and aggregate size increase, *ii)* particles become more organic, and *iii)* more porous, thus $d_f$ decreases. Based on a first order approach, we assume a linear decrease or increase of parameters with distance from coast.  For boundary conditions in coastal waters, we therefore assume an average $\langle D_p \rangle = 4\,\mu\text{m}$

10    due to the dominance of cohesive sediment (Winterwerp, 1998), $\langle D \rangle = 100\,\mu\text{m}$  (Fettweis et al., 2006), $\omega = 0.9$  account for a loss on ignition (LoI) of $\approx 0.04$ (Fettweis, 2008), and $d_f = 2.0$ as an average coastal fractal dimension (Winterwerp, 1998). By contrast, for open waters we presume $\langle D_p \rangle = 10\,\mu\text{m}$ for algae-dominated particles, $\langle D \rangle = 500\,\mu\text{m}$  we assume similar trends as in van der Lee et al. (2009) for the Irish Sea, $\omega = 0.05$

15    to parallel a LoI$\approx 0.89$ (Eisma and Kalf, 1987), and $d_f = 1.6$ as a typical value for fluffy aggregates (Logan and Alldredge, 1989; Li and Logan, 1995). The densities of primary particles are set to $\rho_p = 2650\,\text{kg}\,\text{m}^{-3}$  (Maggi, 2009) and $\rho_o = 1100\,\text{kg}\,\text{m}^{-3}$, which is computed as an average of the range given in Fettweis (2008). The water density and the dynamic viscosity are set to $\rho = 1000\,\text{kg}\,\text{m}^{-3}$ and $\mu = 0.001\,\text{kg}\,\text{m}^{-1}\,\text{s}^{-1}$, respectively. The sensitivity of $w_s$ to changes in each parameter  is assessed by varying each parameter separately while

20    keeping the other  parameters at their typical values for coastal waters, e. g. for an organic-rich aggregate with an assumed diameter of $D = 100\,\mu\text{m}$, and $\omega = 0.5$, $D_p = 4\,\mu\text{m}$, and $d_f = 2$.

**3   Results**

**3.1   Cross-coastal gradients**

Energy dissipation rates $\epsilon$ generally possess spatio-temporal variability. Most relevant for this study, vertically and temporally

25    averaged model-derived $\epsilon$ exhibit a cross-shore gradient from the inner German Bight towards the coast. The temporal averaging  is carried out for the  time of the cruises while accounting for the length of the tidal cycle. Values range between $\epsilon \approx 10^{-9}\,\text{m}^2\,\text{s}^{-3}$ and $\epsilon \approx 10^{-4}\,\text{m}^2\,\text{s}^{-3}$, respectively (Fig. 3).

   Along this range, mean SPMC first  increases from values below $1\,\text{g}\,\text{m}^{-3}$ to approximately $10\,\text{g}\,\text{m}^{-3}$. Above $\log_{10}(\epsilon) = -5.5$,  mean SPMC moderately  decreases to approximately $6\,\text{g}\,\text{m}^{-3}$ and  increases

30    again to $10\,\text{g}\,\text{m}^{-3}$ with further  increase of $\epsilon$ (Fig. 4 A) towards the coast.

   Under the assumption that fluorescence can be applied as a proxy for POM, the potential significance of algae and their products for particle composition is depicted by the mean ratio between measured fluorescence and SPMC. High F/SPMC ratios  indicate rather organic-rich particles, while low F/SPMC ratios  indicate rather mineral particle-dominated flocs. The F/SPMC ratio shows rather high variability for $\log_{10}(\epsilon) < -6$ (Fig. 4 B). With a further increase in $\epsilon$, the ratio drops and levels at approximately a fourth of the open German Bight value for $\log_{10}(\epsilon) > -5$.

Average sinking velocities $\langle w_s \rangle$ show very low values of order $10^{-6}\,\mathrm{m\,s^{-1}}$ to $10^{-5}\,\mathrm{m\,s^{-1}}$ in regions of $\log_{10}(\epsilon) < -7.5$ (Fig. 4 C). At higher $\epsilon$, $\langle w_s \rangle$ increases significantly reaching a maximum of about $7 \cdot 10^{-4}\,\mathrm{m\,s^{-1}}$ around $\log_{10}(\epsilon) \approx -5.5$. This maximum $\langle w_s \rangle$ around $\log_{10}(\epsilon) \approx -5.5$ coincides with an increase of SPMC, and a drop in fluorescence to SPMC ratio. By further increasing $\epsilon$, $\langle w_s \rangle$ decreases again to values of $\langle w_s \rangle \approx 3 \cdot 10^{-4}\,\mathrm{m\,s^{-1}}$. The Monte-Carlo-type simulation to estimate parameter uncertainties exhibits the same pattern, expressed as $\sigma$ and $2\sigma$ confidence levels, 68 % and 95 % respectively (Fig. 4C), around their ensemble mean, and thus underpins the results of $\langle w_s \rangle$ along $\epsilon$.

**3.2 Conceptual model**

Sinking velocity of an average particle (as parametrized in Sec. 2.4) changes almost linearly with variations of single parameters from open to coastal waters (Fig. 5, upper four panels). Flocs would sink less rapid because of the decreasing diameter of the aggregate or primary particles in the coastal region. By contrast, increasing amounts of mineral particles and fractal dimension would both lead to increased $w_s$. As an overall effect, when considering all parameters, $w_s$ reaches a maximum in between the near-shore and open waters (lowest panel of Fig. 5).

**4 Discussion**

**4.1 Sinking velocity on a gradient of prevailing energy dissipation rate**

Sinking velocities were determined along a cross-shore transect in the German Bight defined by the prevailing $\epsilon$. For the reconstruction of $\langle w_s \rangle$, we had to assume congruency between *in situ* measurements and hydrodynamic model results within a range defined by the applied filters. The estimated $\langle w_s \rangle$ of order $10^{-6}\,\mathrm{m\,s^{-1}}$ to $5 \cdot 10^{-4}\,\mathrm{m\,s^{-1}}$ in low and high energy dissipation regions, respectively (Fig. 4), compare very well with former *in situ* studies in the German Bight and adjacent areas. Puls et al. (1995) found $w_s \approx 10^{-6}\,\mathrm{m\,s^{-1}}$ to $2 \cdot 10^{-5}\,\mathrm{m\,s^{-1}}$ for the open German Bight, a low $\epsilon$ region, and Pejrup and Mikkelsen (2010) reported median $w_s$ of $1 \cdot 10^{-4}\,\mathrm{m\,s^{-1}}$ to $11 \cdot 10^{-4}\,\mathrm{m\,s^{-1}}$ for the Rømø and Høyer Dyb, Danish Wadden Sea where $\epsilon$ is comparably high. Our estimated $\langle w_s \rangle$ are also well in the range found in other regions, e. g. in Chesapeake Bay (Gibbs, 1985) or at the Belgian Coast (Fettweis and Baeye, 2015). In agreement with former studies (e. g. compiled in Dyer, 1989), $\langle w_s \rangle$ increases with increasing $\langle \mathrm{SPMC} \rangle$ (Fig. 6), which gives further confidence in the methodological approach. However, the correlation is rather poor compared to former local studies and can be explained by the heterogeneity of the German Bight system, seasonal effects and the intrinsically high variability of sinking velocities (Fettweis, 2008) and of turbulence (Pejrup and Mikkelsen, 2010). The significant maximum of $\langle w_s \rangle$ at $\langle \epsilon \rangle \approx 10^{-5.5}\,\mathrm{m^2\,s^{-3}}$ can be explained by the balance between aggregation and fragmentation. Both processes are controlled by turbulent shear that is generated by energy dissipation, SPM volume concentration, and adhesion properties of particles involved. Previous flocculation modeling studies point to a formation of $w_s$ maximum along $\epsilon$ (Winterwerp, 1998; Pejrup and Mikkelsen, 2010). Our findings of a maximum $\langle w_s \rangle$ between

$\epsilon \approx 10^{-6}\,\mathrm{m}^2\,\mathrm{s}^{-3}$ and $10^{-5}\,\mathrm{m}^2\,\mathrm{s}^{-3}$, which translates to shear rates of about $1\,\mathrm{s}^{-1}$  $3.2\,\mathrm{s}^{-1}$ for $\nu = 10^{-6}\,\mathrm{m}^2\,\mathrm{s}^{-1}$, should

thus be comparable to former theoretical studies. For example, in a field work-based modeling approach Pejrup and Mikkelsen

5 (2010) found the maximum of sinking velocity to occur under higher shear rates of about $8.5\,\mathrm{s}^{-1}$. By contrast, Maerz et al.

(2011) calculated highest mean sinking velocities of about $5 \cdot 10^{-4}\,\mathrm{m}\,\mathrm{s}^{-1}$, similar to our findings, for a jar test device-based

laboratory experiment with natural SPM during shear rates of about $1\,\mathrm{s}^{-1}$. Given the underlying uncertainties in calculating

the shear rate in both cases, our study  agrees well with former theoretical studies. Hence, as  suggested by Dyer

(1989) and Pejrup and Mikkelsen (2010), in addition to SPMC, turbulent shear can be regarded as the major determinant for

10 $w_s$.

**4.2 Sinking velocity as a result of SPMC, composition and a turbulence gradient**

Our study additionally suggests a change in particle composition with $\epsilon$ as proxied by the  fluorescence-to-SPMC

ratio. With $\epsilon$, SPMC increases towards the coast and mineral fraction becomes dominant compared to organic particles at low $\epsilon$

(Eisma and Kalf, 1987). Hence, density and  other physico-chemical properties of primary particles such as size and ad-

15 hesion can change accordingly and thus influence $w_s$ along a cross-shore transect. Our conceptual model illustrates the effect of

varying particle properties on $w_s$ independent of the observations used in our data analysis. However, the course of the parameters implicitly presumes certain environmental conditions,  such as particle size being, among others,

a function of SPMC and shear. Single parameter changes in the conceptual model  result in mostly linear responses in

$w_s$, while considering all parameters  lead to a maximum in $w_s$ on the conceptual cross-shore transect, in alignment with

20 our findings in the data analysis. The assumption of linearly changing parameters is a first order approximation.

Different and non-linear  changes of parameters would lead to a transformation of the maximum sinking velocity zone

such as a shifting and / or stretching (not shown). Former model studies  show that for a given shear regime ,

varying particle properties such as particle adhesion  can lead to a change in particle size and thus

sinking velocity (e. g. Maerz et al., 2011). Hence, higher particle adhesion would translate to larger particles and stronger frag-

25 mentation resistance, which would allow particles to sink out of the water column closer to the coast under higher turbulence

 intensity. This implies that spatial changes in SPMC and composition, and accordingly physico-chemical properties

of particles, potentially affect the location of the transition zone. As discussed below this probably happens on a seasonal time

scale due to the modulation of the transition zone by phytoplankton exudations which requires further investigations. While

the cross-shore distribution of $w_s$ is predominantly controlled by prevailing shear, the conceptual model indicates a potential

30 additional relevance of physico-chemical properties for the formation of the high sinking velocity zone.

**4.3 Implications of a cross-coastal maximum of sinking velocity for biogeochemical cycling in the coastal zone**

The region of $\log_{10}(\epsilon) \approx -5.5$ with highest $\langle w_s \rangle$ coincides with a zone located off the coast at depth of about $15\,\mathrm{m}$ to $20\,\mathrm{m}$.

It is accompanied by a strong gradient in SPMC and  SPM composition that is depicted by the F/SPMC

ratio. This suggests that the region can be considered as a transition zone, hindering mineral particles to escape further offshore. To simplify, consider the course from the coast to open waters. Once formed, relatively dense  fast sinking flocs, whose

[revised manuscript text omitted]

energy dissipation rates. Even though we  neither resolve tidal cycles nor consider potential seasonal variation of $\langle w_s \rangle$

along $\langle \epsilon \rangle$, a map of $\langle w_s \rangle$ allows for  insights into the spatial distribution and variability of $\langle w_s \rangle$. It  is re-

5    constructed by mapping the found bin-wise relationship between $\langle \epsilon \rangle$ and $\langle w_s \rangle$ (Fig. 4C) onto the respective spatial $\langle \epsilon \rangle$ (Fig. 3).

It must be highlighted that local, short term *in situ* measurements would most likely deviate from the obtained averaged

picture (Fig. 7) that would be challenging to derive by *in situ* measurements. Nevertheless, strong spatial variability is visible

with a pronounced  cross-shore gradient of $\langle w_s \rangle$ featuring a maximum of sinking velocities along the coastline

as indicated before. However, this maximum varies locally and is particularly less pronounced at the northern German and

10    southern Danish coast, where $\langle w_s \rangle$ declines to values that are  in the range of $\langle w_s \rangle$ in deeper waters in the southern

German Bight. Applying the aforementioned concept of the transition zone as an off-coastal closing mechanism for nutrient

cycling to the two contrasting German Wadden sea regions, East Frisian Wadden Sea and North Frisian Wadden Sea, in the latter

particularly the Sylt-Rømø Bight, may help to better understand regional differences of nutrient concentrations. Lower nutrient

concentrations and thus lower eutrophication levels in the Sylt-Rømø bight compared to the East Frisian Wadden sea are

15    regularly observed (van Beusekom et al., 2009). In this case, the maximal $\langle w_s \rangle$ calculated in front of the Sylt-Rømø tidal inlet

 amounts only to $4 \cdot 10^{-4}\,\mathrm{m\,s^{-1}}$, about half the magnitude found in front of the other Wadden Sea regions. Hence, the ability

for nutrient retention is diminished and would contribute to the lower nutrient concentrations compared to other Wadden Sea

regions. By contrast, the potentially pronounced retention capacity of the transition zone, as predicted for the Dutch coast and

German East-Frisian islands may contribute to the phenomenologically described *line of no return* (Postma, 1984), expected

20    further off the coast and characterized by low SPMC. The spatial distribution of $\langle w_s \rangle$ is probably also reflected in the SPMC

and their cross-coastal gradients (Fig. 8). Flocs with high $w_s$  are likely mineral-dominated  and would rapidly sink out

of the water column and thus lead to strong cross-coastal diminishing of SPMC. Generally, dilution of SPMC occurs due to

cross-shore wise increasing water depth and  local currents parallel to the coast potentially  confining

horizontal SPMC distribution to the near-coast region (Staneva et al., 2009). Nevertheless, on average, lower $\epsilon$ accompanied by

25    lower SPMC and smaller horizontal gradients are found along the North-Frisian than the Dutch and East-Frisian coast (Fig. 3

and 8). In combination, these conditions, in particular along the Sylt-Rømø islands, suppress the establishing of a defined

transition zone.  Hence, lower $w_s$ should be found as  implied by Fig. 7. As a result, retention efficiency is

likely reduced  and potentially contributes to the observed lower nutrients concentrations compared to the East-Frisian

Wadden Sea (van Beusekom et al., 2009).

30    Other concepts  are presented to explain the observed spatial heterogeneity of eutrophication levels along the Wadden

Sea coast. These explanations  are categorized by Van Beusekom et al. (2012) in two main groups: either due to i) regional

differences in organic matter import, or ii) differences in the size of the tidal basins. Potential differences of the import amount

were attributed to different i) regional offshore primary production, ii) orientation of the coastline with respect to dominant

wave and current directions and iii)  intensity of shore-ward bottom currents. The eutrophication status can be related

35    to the tidal basin width, where narrow tidal basins feature higher eutrophication status than wider ones (Van Beusekom et al.,

2012). While the general build-up of nutrients gradients towards the coast  is attributed to the above discussed processes

of *settling and scour lag* in combination with the estuarine circulation, no clear mechanistic explanation  is drawn

as to why the size of the tidal basins lead to the found relation. Van Beusekom et al. (2012) suggested that POM imported into the tidal basins would be either distributed over a larger area for wide basins which would lead to gentle nutrient gradients compared to steeper nutrient gradients in narrower basins. However, other explanations cannot be ruled out, e. g. linked to differences in water exchange times (Van Beusekom et al., 2012). It is likely that a multitude of processes interact and detailed modeling studies are required to disentangle their relative contributions to the observed spatial heterogeneity of the Wadden Sea eutrophication status. Noticeably, the Sylt basin  exhibits exceptionally low eutrophication status in the study of Van Beusekom et al. (2012), which would fit into the picture that the import of POM is reduced due to the rather weakly established transition zone in this area.

Since the driving mechanisms behind the transition zone are probably ubiquitous for coastal marine systems with sufficiently strong cross-shore water density gradients generated by freshwater run-off, precipitation-evaporation balance or heat fluxes, local hydrodynamics will determine its eventual formation. This implies that the concept of the transition zone with its retention capacity for nutrients may be applicable to other coastal marine and estuarine systems in general, and will help to better understand different nutrient cycling behavior among those systems.

**4.5 Biological modulation of the transition zone?**

There is increasing evidence that sticky algae excretions such as transparent exopolymer particles mediate aggregation and increase shear resistance of particles against fragmentation, potentially enabling algae to clear the water column from mineral load (Fettweis et al., 2014). As shown in the conceptual model (Sec. 2.4), high $w_s$ occur due to the interplay of shear-driven flocculation and primary particle density and size. Similarly, Hamm (2002) found a non-linear effect of mineral-organic weight/weight-content on sinking velocity showing first an increasing $w_s$ for increasing mineral load, but a stagnation or even converse effect when exceeding a certain threshold. To hypothesize, a high concentration of SPM with an optimal ratio between dense mineral and sticky bio-particles might exist in the transition zone under  favorable turbulent conditions to form larger flocs compared to near-coast regions, and denser flocs compared to open waters. In sum, this could lead to the found high $\langle w_s \rangle$. This raises the important question as to what extent algae can affect the transition zone and its seasonal location to sustain the effective coastal nutrient cycle that is driven by the estuarine circulation. By sustaining the nutrient cycle, algae  potentially support the development of the pronounced nutrient gradients towards the Wadden Sea. As a consequence, 3-D biogeochemical models require a representation of the tight coupling between sediment dynamics and biogeochemical, potentially even adaptive phytoplankton physiological processes to improve models' capability to estimate particulate matter fluxes.

**5 Conclusions**

The present study provides a strong indication of a maximum of sinking velocities along a cross-shore transect that is defined by a gradient of prevailing energy dissipation rates. Towards the off-shore, the enhanced sinking velocities are accompanied by

a strong decline of SPMC, and an increasing F/SPMC ratio.  The interplay of processes leading to the observed features defines the region as coastal transition zone. In turn, processes feed back on SPMC and composition.

The transition zone is probably an important feature on the course from the coast to the continental shelf. Predominantly driven by turbulent shear generated by energy dissipation, the transition zone with highest sinking velocities potentially acts as a retention zone for dissolved nutrients leaking from near-shore waters by providing favorable conditions for algae growth. Phytoplankton take up nutrients and excrete extracellular polymeric substances (EPS) that mediate flocculation processes. Embedding minerals into the organic matrix leads to enhanced sinking velocities. Algae thus seem to possess the ability to clear the water column (Fettweis et al., 2014), and link the off-shore dissolved nutrient fluxes to residual landward bottom fluxes. Consequently, these interlinked processes would have the potential to retain fine sediments and nutrients in coastal areas. The strength of the links eventually affects the eutrophication state of the coastal region.

It is scarcely understood  what relevance individual processes have in forming the transition zone. Besides  favoring energy dissipation rates of about $\epsilon \approx 10^{-5.5}\,\mathrm{m^2\,s^{-3}}$, algae and their extracellular polymeric excretions are potentially  important in forming of the transition zone as EPS are known to enhance collision efficiency and resistance against fragmentation. Since  primary production in the studied area features a pronounced seasonal cycle, further studies are thus needed to investigate the temporal and spatial extension of the transition zone. Long term or repeated spatial *in situ* measurements, including SPMC, $\epsilon$ profiles and particle properties such as size, $w_s$ and (volume-) composition, are required to underpin the indirectly found enhanced sinking velocities as a feature of the transition zone. Additionally, modeling approaches are required to gain a deeper process-based understanding, and to disentangle the closely linked processes among other eutrophication state affecting environmental factors. This also implies the necessity to incorporate biological-mineral interactions in models, especially  when system-wide studies are carried out.

*Author contributions.* R. Riethmüller planed and carried out the ScanFish measurements in the German Bight and took responsibility to ensure data quality. E. M. van der Lee conducted an intensive pre-analysis of the data. U. Gräwe provided hydrodynamical model results from GETM for the times of the cruises. R. Hofmeister handled the 3D GETM results to transfer them to the ScanFish track. J. Maerz and K.W. Wirtz discussed and outlined the general research approach. J. Maerz carried out the data analyses and prepared the manuscript. All co-authors critically contributed to the discussion and to the interpretation of the results and the compilation of the manuscript.

*Acknowledgements.* The authors thank the crew of the research vessel RV Heincke, H. Rink, H. Thomas, M. Heineke, and R. Kopetzky for operating the ScanFish, for processing and quality assurance of the observational data and laboratory work, and L. Merckelbach and G. Flöser for fruitful discussions. We would like to thank the two anonymous reviewers who helped to improve our manuscript. This work was supported *i)* by the German Federal Ministry of Research and Education in the framework of the projects PACE (The future of the Wadden Sea sediment fluxes: still keeping pace with sea level rise?, FKZ 030634A) and MOSSCO (Modular System for Shelves and Coasts, FKZ 03F0667A), *ii)* by the Helmholtz society via the program PACES, *iii)* by the Niedersächsisches Ministerium für Wissenschaft und Kultur (MWK) and the Niedersächsisches Ministerium für Umwelt und Klimaschutz (MUK) in the framework of the WiMo project (Scientific monitoring concepts for the German Bight), and through the Coastal Observing System for Northern and Arctic Seas (COSYNA). The

financing of further developments of the IOW's Baltic Monitoring Program and adaptations of numerical models (STB-MODAT) by the federal state government of Mecklenburg-Vorpommern is greatly acknowledged by Ulf Gräwe. Supercomputing was provided by the North-German Supercomputing Alliance (HLRN). Observational and model data and related material are available for scientific purpose upon request to the corresponding author.

[revised manuscript text omitted]